# Clonal analysis and dynamic imaging identify multipotency of individual *Gallus gallus* caudal hindbrain neural crest cells toward cardiac and enteric fates

Weiyi Tang [1], Yuwei Li[1], Ang Li[2] & Marianne E. Bronner[1]✉

Neural crest stem cells arising from caudal hindbrain (often called cardiac and posterior vagal neural crest) migrate long distances to form cell types as diverse as heart muscle and enteric ganglia, abnormalities of which lead to common congenital birth defects. Here, we explore whether individual caudal hindbrain neural crest precursors are multipotent or predetermined toward these particular fates and destinations. To this end, we perform lineage tracing of chick neural crest cells at single-cell resolution using two complementary approaches: retrovirally mediated multiplex clonal analysis and single-cell photoconversion. Both methods show that the majority of these neural crest precursors are multipotent with many clones producing mesenchymal as well as neuronal derivatives. Time-lapse imaging demonstrates that sister cells can migrate in distinct directions, suggesting stochasticity in choice of migration path. Perturbation experiments further identify guidance cues acting on cells in the pharyngeal junction that can influence this choice; loss of *CXCR4* signaling results in failure to migrate to the heart but no influence on migration toward the foregut, whereas loss of *RET* signaling does the opposite. Taken together, the results suggest that environmental influences rather than intrinsic information govern cell fate choice of multipotent caudal hindbrain neural crest cells.

[1] Division of Biology and Biological Engineering, California Institute of Technology, Pasadena, CA 91125, USA. [2] Department of Kinesiology, College of Nursing and Health Innovation, University of Texas at Arlington, Arlington, TX 76019, USA. ✉email: mbronner@caltech.edu

A challenge in understanding cell fate choice is to identify how individual embryonic stem cells make lineage decisions in order to differentiate into diverse adult cell types. A remarkable example of such a stem cell population is the neural crest, which forms multiple derivatives ranging from peripheral neurons and glia to pigment cells, cartilage, and muscle cells[1]. A long-standing question is whether individual neural crest cells are multipotent stem cells or fate-restricted progenitors, each "pre-determined" to form a particular cell type. While this question has been elegantly tackled for trunk neural crest cells in the mouse using Confetti technology[2], lineage analysis at other levels of the neuraxis has been more challenging due to their vast array of derivatives coupled with long distances traveled. The question of multipotency is particularly pertinent to neural crest cells arising from the caudal hindbrain, which undergo some of the longest migrations of any embryonic cell type and populate derivatives as diverse as muscle of the heart[3,4], and neurons and glia of the sympathetic and enteric nervous system (ENS)[5].

The caudal hindbrain neural crest arises adjacent to the middle of the otic vesicle to the level of somite 7 and is termed "vagal" neural crest. Transplantation and cell labeling techniques have shown that the vagal region can be subdivided into two populations. Cells arising from adjacent to the mid-otic vesicle to somite 3, corresponding to rhombomeres (r) 5 and 6, are called the "cardiac" neural crest since many of these cells populate the heart, contributing to the aorticopulmonary septum, cardiac ganglia, pharyngeal arch arteries, cardiomyocytes as well as parts of the thymus, thyroid, and parathyroid[3,4,6]. In contrast, the "posterior vagal" neural crest arising adjacent to somites 4-7 develops into ganglia of the sympathetic and enteric nervous systems[5].

Grafting experiments at the population level have suggested differences in the developmental potential of vagal neural crest cells depending upon their rostrocaudal level of origin. For example, when small grafts of quail neural folds were transplanted into chick hosts from which the vagal neural crest was removed, cells adjacent to somite 3 could form enteric neurons whereas those from adjacent to somite 1 failed to do so[7]. Similarly, GFP labeling via somite level-specific electroporation or transplantation in chicken embryos has shown that while cardiac neural crest cells migrate to the heart and the gut, posterior vagal neural crest cells contribute only to the ENS[8,9]. However, disagreement remains on the nature of the migratory pathways and cell types emerging from the cardiac neural crest region. Whereas Kuo and Erickson suggest that cardiac neural crest cells take different pathways to the heart than to the gut[8], Espinosa and colleagues highlighted the role of vagus nerve in guiding Schwann cell precursors migration toward the ENS[9]. These population level analyses clearly show that cells delaminating from the cardiac level can contribute to both cardiovascular system and the ENS, albeit the source of the cells, whether of neural crest and/or Schwann cell precursor origin, remains a matter of debate[8,9].

These studies raise the intriguing question of whether individual neural crest cells are multipotent, with the potential to form both cardiac and enteric derivatives, or if there are distinct precursors fated to populate each of these distant destinations that migrate differentially. Addressing this question requires examining the developmental potential of individual cardiac neural crest precursor cells. This has been done in vitro by isolating cardiac neural crest cells at migratory[10] and post-migratory[11] stages. In these clonal cultures, some individual neural crest-derived cells were capable of generating multiple cell fates, including pigment cells, smooth muscle cells, connective tissue cells, chondrocytes, and sensory neurons. Further studies showed that the scope of developmental potential is partially restricted in *TRKC* mutant mice[12], suggesting that environmental factors may play a role in maintaining stem cell properties of the cardiac neural crest. While

these experiments have tested the potential of cardiac neural crest cells removed from the embryonic milieu and grown at clonal density, it is technically difficult to extend these analyses in vivo.

To meet this challenge, here we perform clonal analysis of the cardiac neural crest in vivo at high resolution. We first use replication-incompetent avian (RIA) retroviruses[13–18] that express distinct fluorophores to identify clonally related progeny of premigratory cardiac and posterior vagal neural crest cells. Next, we establish an assay for anterograde lineage tracing of migratory cells. Through sparse viral infection and photoconversion using a one-photon laser, we can precisely photolabel single migrating cells within the complex context of an embryo. Finally, by coupling molecular perturbations with our recently established dynamic imaging approach[15,19], we show that early cardiac neural crest cells migrate stochastically; once within the pharyngeal junction, they migrate toward both the heart and gut as a result of target-specific responses to *CXCR4* and *RET* signaling. While posterior vagal neural crest cells are also multipotent, they contrast with cardiac crest cells by only contributing to ENS and sympathetic ganglia. Taken together, our study identifies interesting similarities and differences between individual neural crest precursors emerging from distinct levels of the caudal hindbrain. While both cardiac and posterior vagal neural crest cells are multipotent, they undergo differential migration. Cardiac neural crest cells can populate both the heart and gut under the influence of differential guidance cues, whereas posterior vagal neural crest cells migrate to the gut in a manner apparently independent of these guidance cues.

## Results

**Multiplex clonal analysis shows that premigratory neural crest cells at the "cardiac" level are multipotent.** At Hamburger Hamilton (HH) stage 9, cardiac neural crest precursors are positioned within the dorsal neural tube and about to delaminate. A mixture of RIA retroviruses encoding distinct fluorescent proteins (Fig. 1a) was injected into the neural tube from the mid-otic to somite 3 level and embryos were harvested 48 h post infection (Supplementary Fig. 1a, b show examples of embryos immediately after and 48 h post-injection). A slice was cut directly posterior to the otic placode (Fig. 1b) to obtain a transverse view of all cardiac neural crest-related structures (Fig. 1c, gray), including the dorsal neural tube (dNT), cranial nerve nine (CN-IX), pharyngeal arch arteries (PAA), and outflow tract (OFT). The tissue slice was imaged live in order to preserve native fluorescence signal without compromising normal biological processes.

To determine the probability of multiple infection and thus assess clonality, we performed a calculation similar to that used for trunk neural crest precursors[16,20] but revised to account for the increased numbers of Pax7+ cells in the caudal hindbrain compared with the trunk neural tube. After estimating the numbers of Pax7+ versus total number of cells of the hindbrain, we calculated the probability of multiple infections $P\{n\}$ using the following equation:

$$P\{n\} = \frac{m^n e^{-m}}{n!} \tag{1}$$

where n is the number of viral particles simultaneously infecting a premigratory neural crest cell (Pax7 + ); m is average number of viral particles per Pax7+ cell (Supplementary Fig. 1c, d). To determine m, we measured the titer of the virus which was ~1 × $10^7$ pfu/ml, in the volume injected into the embryo, which was typically ~0.5 µl, corresponding to ~5000 particles injected. Of these viral particles, the proportion infecting Pax7+ cells is equal to the ratio between the number of Pax7+ cells and total cells in the hindbrain neural tube (~0.258). Thus, the number of viral

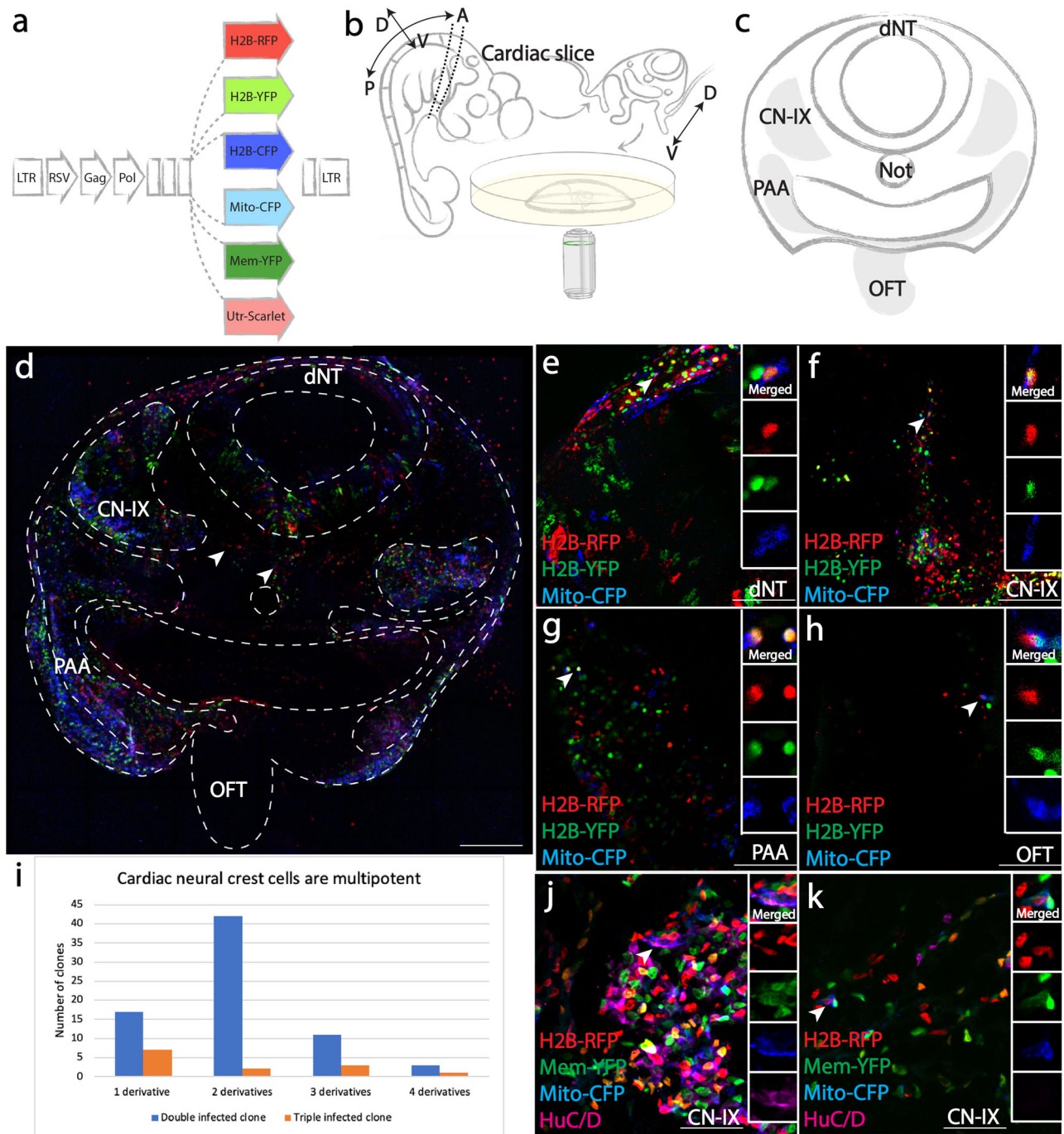

**Fig. 1 Multiplexed clonal analysis demonstrates multipotency of premigratory cardiac neural crest cells. a** Sequences encoding H2B-RFP, H2B-YFP, H2B-CFP, Mito-CFP, Mem-YFP, and Utrophin-Scarlet were cloned downstream of the RSV promoter of the RIA vector for combinatorial infection. **b** Schematic diagram of in vivo slice culture. At HH21, a ~500 μm thick slice was obtained posterior to the otic placode. The slice was then embedded in 1% agarose and imaged using inverted confocal microscopy. A, anterior; P, posterior; D, dorsal; V, ventral. **c** Schematic diagram of cardiac neural crest-derived tissues (gray). Dorsal neural tube (dNT), cranial nerve nine (CN-IX), pharyngeal arch arteries (PAA) and outflow tract (OFT) were imaged and scored for cells with identical clonal signatures. **d** A tiled multiplex image of the cardiac slice, representative of 23/23 embryos. CN-IX, PAA, OFT and Schwann cells (arrowhead) were infected by a mixture of RIAs. Red signal along the pharyngeal endoderm is autofluorescence from tissue debris common in chick explant cultures. **e–h** Clonally related cells (ID: Supplementary Data 1a-84) infected by three viruses (H2B-RFP, H2B-YFP, Mito-CFP, arrowhead) were identified in all four cardiac neural crest derivatives: dNT (**e**), CN-IX (**f**), PAA (**g**), OFT (**h**). In (**h**), RFP and YFP signals did not align in the OFT as a result of heart beating during image acquisition. **i** Bar graph showing distribution of double (blue) and triple (orange) infected clones. **j, k** In clones that appeared to be restricted in one location, cells could acquire distinct fates. At E7, one cell in a rare clone near CN-IX expressing H2B-RFP, Mem-YFP and Mito-CFP differentiated into HuC/D expressing neuron (**j**), while another cell of the same clone remained HuC/D negative (**k**). Scale bars: **d** 500 μm; **e–h** 100 μm; **j, k** 60 μm.

particles infecting Pax7+ cells is around 1292. The number of Pax7+ cells (~5863 cells) in the caudal hindbrain was estimated based on average number of cells in each section and thickness of the caudal hindbrain segment; m is therefore around 0.22. Collectively, this led to the conclusion that the probability of a single cell being simultaneously infected by two or more viruses was low (0.019 for double infection and 0.0014 for triple infection)[16,20]. Thus, in both double and triple infections, the unique fluorescence signatures from multiple infection events can be utilized as clonal readouts. To further improve the stringency of our clonal analysis, we defined a clone by three criteria: a rare complex color combination produced by two or more infections of an individual precursor (multiple infection); similar intensity of fluorescent signal (due to site of integration within the genome); subcellular localizations of fluorescent proteins (Fig. 1a, e.g., nuclear, mitochondrial, membrane, or actin cortex).

Upon visual inspection of living tissue, fluorescence was detected in all cardiac neural crest derived regions (Fig. 1d). Zooming in on neural tube and neural crest derivatives, clones with identical color/subcellular signatures were identified in multiple structures. For example, a clone simultaneously expressing H2B-RFP, H2B-YFP and Mito-CFP shared progeny in the dNT (Fig. 1e), CN-IX (Fig. 1f), PAA (Fig. 1g), and OFT (Fig. 1h). In general, neural crest-derived clones displayed variable sizes, with an average of 9 cells per clone, indicating diverse proliferative properties. 18/86 (21%) of these clones developed into two or more neural crest derivatives in addition to the dNT and 4/86 (4.6%) of the clones exhibited an even higher degree of multipotency with daughter cells residing in all three neural crest derivatives (Fig. 1i, Supplementary Data 1a). In contrast, 24/86 (28%) clones gave rise to progeny within a single location.

Immunofluorescence analysis showed that clonally related cells can adopt multiple cell fates even within a single derivative. For example, descendants of a rare clone expressing H2B-RFP, Mem-YFP and Mito-CFP were only seen within CN-IX. However, staining with the neuronal marker HuC/D at embryonic day 7 (E7), a time after neuronal differentiation, identified one cell at the center of CN-IX that expressed the neuronal marker HuC/D (Fig. 1j), but another HuC/D negative cell of the same clone at the periphery of the ganglion, a location where presumptive glial cells often reside, consistent with a presumptive non-neuronal cell fate (Fig. 1k). These results suggest that individual premigratory cardiac neural crest cells can produce diverse cell types even when in close proximity, indicating that location alone may underestimate the developmental potential within a clone.

**Single-cell photoconversion shows that migratory "cardiac" neural crests are multipotent.** Next, we asked whether cardiac neural crest cells retain multipotency during dorsoventral migration. As these cells move toward the heart, it becomes difficult to utilize multiplex viral labeling owing to increasing tissue depth and the large number of virally infected cells. We, therefore, developed a technique, comprised of sparse viral infection followed by one-photon laser photoconversion, to label a single migrating cell with exquisite spatiotemporal control.

To this end, we constructed a recombinant RIA virus encoding H2B-Emeos that exhibits high levels of photoconversion efficiency and fluorescence intensity. Previously, two methods for single-cell photoconversion have relied on confined axial excitation of cells expressing photoactivatable or photoconvertible proteins. While the femtosecond pulsed two-photon laser serves as the gold standard for this purpose because of its low scattering near-infrared light confines the excitation[21], several studies have reported that negligible photoconversion was achieved even using high-power illumination and long-term exposure. Primed

conversion, in which simultaneous illumination of the specimen with both 488 nm and 730 nm lasers, can also achieve single-cell photoconversion[22]. However, the requirement of additional hardware (a filter cube to transmit both lasers) and the use of high-power infrared laser (730 nm) preclude its wide application. Therefore, we addressed these issues by manipulating viral titer to sparsely tag neural crest cells, which can be efficiently converted through the non-confined 405 nm one-photon laser, thus simplifying the tools for clonal analysis. While used here for clonal analysis of the neural crest, this could easily be adapted to diverse embryonic tissues, model organisms, and adult stem cells.

Using low viral titers, we achieved sparse labeling of cardiac neural crest cells. A single labeled cell was identified in the optical path, and converted from green to red using 405 nm laser illumination. This approach provides an efficient way to spatially single out an individual progenitor cell for lineage tracing. To distinguish the converted red neural crest cells from blood cells that emit red autofluorescence (Supplementary Fig. 1e, f), we used a 2A bicistronic system to incorporate YFP-Utrophin (targeting the cortex beneath the plasma membrane) in the same virus (Fig. 2a 1). We injected recombinant virus into the neural tube and focused on a region with a single migratory cell along the imaging axis (Fig. 2 a 2, 2′). We then photoconverted the nucleus from green to red while the cell membrane remained green in color (YFP-Utrophin) (Fig. 2 a 3, 3′, b, b′ in explant culture). This specific fluorescence signature permits accurate tracking of the labeled progeny cells during normal development. Despite dilution after cell division, the signal was detectable by confocal imaging at 24 h after tissue fixation and immunohistochemistry with anti-HNK1, a molecular marker for migrating neural crest cells (Fig. 2a 4, c, c′ in cryosections).

Whereas some sister cells remained in close proximity after cell division (3/11), in the majority of cases (8/11), clonally related cells were dispersed in multiple cardiac neural crest derivatives (Supplementary Fig. 1g). For instance, we observed bipotent clones with sister cells in cranial nerve nine (CN, Fig. 2d d1) and pharyngeal arch arteries (PAA, Fig. 2d d2) as well as clones with cells in both pharyngeal arch arteries (PAA, Fig. 2e e1) and the outflow tract (OFT, Fig. 2e e2). Thus, this approach confirms multipotency of individual cardiac neural crest cells in a manner independent of multiple infection and highlights the fact that neural crest multipotency is retained at the migratory stage.

**Premigratory and migratory neural crest cells at the "posterior vagal" level are multipotent.** Having demonstrated the utility of multiplex clonal analysis and single-cell photoconversion at the cardiac level, we applied both approaches to explore clonal relationships within the posterior vagal neural crest region. The same mixture of RIA retroviruses (Fig. 1a) was injected into the entire vagal neural tube, adjacent to somites 1-7, prior to cell delamination. 48 h post-infection, a tissue slice was cut between the levels of somite 4 and 7 (Fig. 3a). Rare color combinations could be identified in four locations including dNT, dorsal root ganglia (DRG), sympathetic chain (SYM), and enteric nervous system (ENS) (Fig. 3b, gray). For instance, a clone expressing Utrophin-Scarlet, H2B-YFP and Mito-CFP was observed in all four locations (Fig. 3d–g). In contrast to the cardiac crest, ~40% of posterior vagal clones (38/95) were confined to a single derivative. We noted that 9/38 (24%) of these "single derivative clones" were located only in the ENS, the most ventral derivative along the migration route (Supplementary Data 1b), without sister cells in the neural tube or the sympathetic chain adjacent to somites 4-7. However, a large number of clones (36/95, 38%) gave rise to progeny in three or four derivatives (Fig. 3c, Supplementary Data 1b). As further validation of the multipotency of some

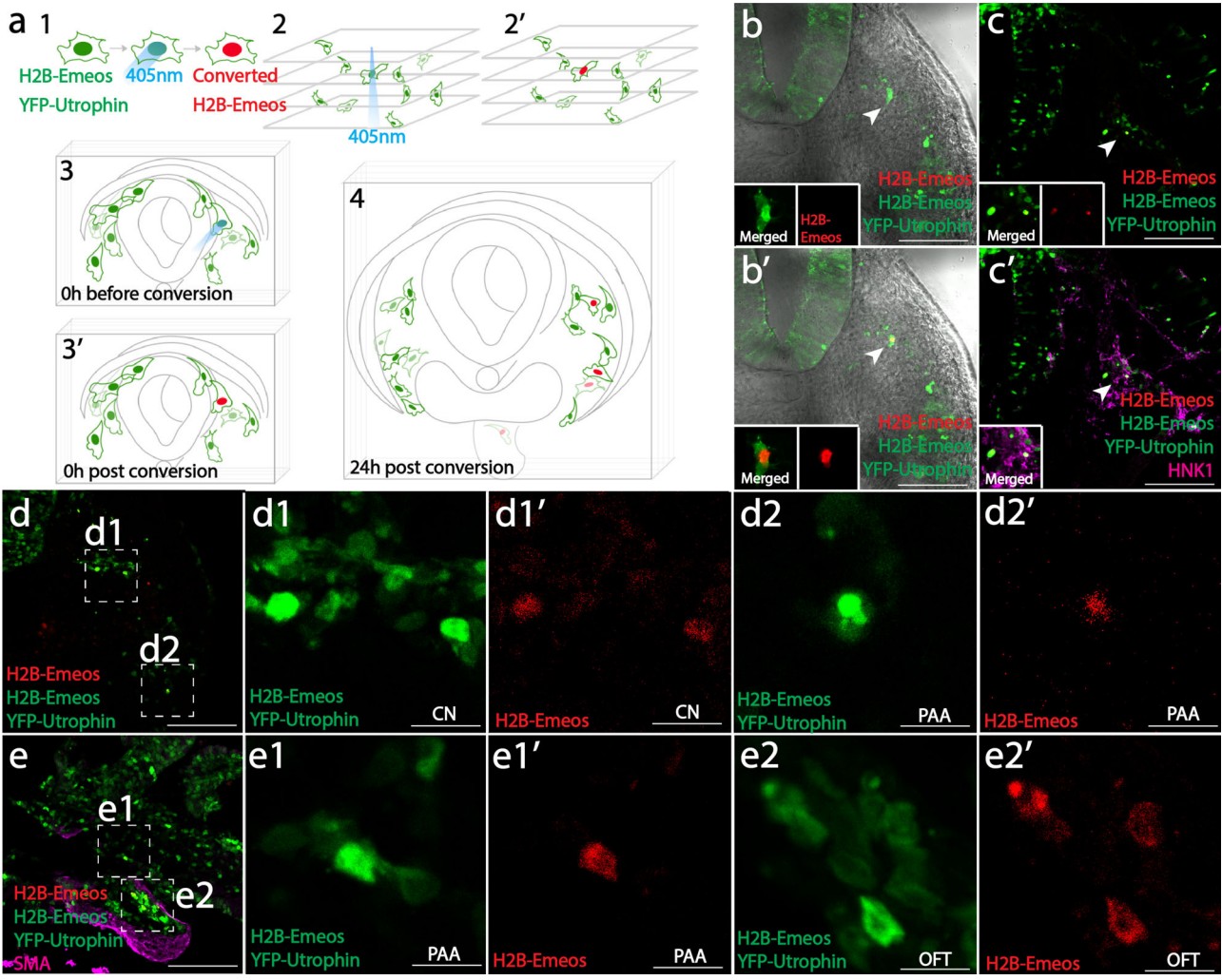

**Fig. 2 Single-cell photoconversion identifies multipotency of migratory cardiac neural crest cells. a** Implementing the single-cell photoconversion technique in embryo slices. In (1), cells were labeled with virus expressing both YFP-Utrophin and H2B-Emeos. A single migratory neural crest cell was converted (2, 2′, 3, 3′), and the tissue slice was cultured for 24 h until its progeny migrated to their destinations (4). Clones are assessed based on red fluorescence in the nucleus and green fluorescence underneath the plasma membrane. **b, c** A representative experiment of single-cell photoconversion and lineage tracing. One migratory cell was photoconverted (b, b′); after 24 h, two daughter cells were observed in the migration stream (c), where cranial nerve is expected to form. Both cells were positive for neural crest marker HNK1 (c′, magenta). **d** An example of a clone with two cells in the cranial nerve (inset d1, CN), and the other cell in prospective pharyngeal arch arteries (Inset d2, PAA). **e** An example of a clone with one cell located in pharyngeal arch arteries (Inset e1, PAA), and three cells migrating further into the outflow tract (Inset e2, OFT). Also see Supplementary Fig. 1e, f for signal detection method and summary of the results (Supplementary Fig. 1g). Scale bars: **b**–**e** 200 μm; Insets d1, d2, e1, e2 12.5 μm.

posterior vagal neural crest cells at migratory stages, we optically highlighted a single migratory posterior vagal crest cell (Fig. 3h), and confirmed that it was able to give rise to both sympathetic cells (Fig. 3i) and part of the enteric nervous system (Fig. 3j).

The fact that so many clones contributed only to the ENS raised two possibilities. First, ENS precursors from the posterior vagal stream may be 'set aside' prior to emigration. Alternatively, due to an overlapping contribution of the cardiac neural crest and posterior vagal neural crest populations to the gut[8,9], these apparently "single-derivative clones" in the ENS might share clonal origin with the more anterior cardiac neural crest rather than being solely derived from posterior vagal neural crest.

**Cardiac neural crest origin of some enteric neurons.** To distinguish between these two possibilities, we injected the RIA viral mixture into the lumen of the neural tube specifically at the cardiac neural crest (mid-otic to somite 3) axial level and analyzed slices at both the cardiac and posterior vagal crest regions

(Fig. 3k). For clonal analysis, four sites including the dorsal neural tube (dNT), cardiac neural crest derivatives (CN-IX, PAA, OFT, Fig. 1c) and the ENS (Fig. 3b) were imaged to enable assessment of both the dorsal-ventral and anterior-posterior extent of the clones. We first screened for specificity of the initial viral infection by examining the posterior vagal neural tube, where we found no fluorescently labeled cells (Fig. 3k).

Importantly, we noted that a considerable number of ENS cells shared clonal signatures with sister cells in cardiac crest derivatives. For instance, we identified a triple labeled clone expressing H2B-RFP, H2B-YFP and Mem-YFP with cells in the dNT at the cardiac crest level (Fig. 3l), CN-IX (Fig. 3m), PAA (Fig. 3n) and ENS (Fig. 3o). In addition, cells in the ENS were clonally related to other cardiac crest derivatives in 24/25 (96%) of cases. 25/72 (35%) of all clones in the cardiac crest shared common progenitor with the gut (Supplementary Data 1c). These results show that a significant proportion of cardiac crest clones also contribute to the ENS.

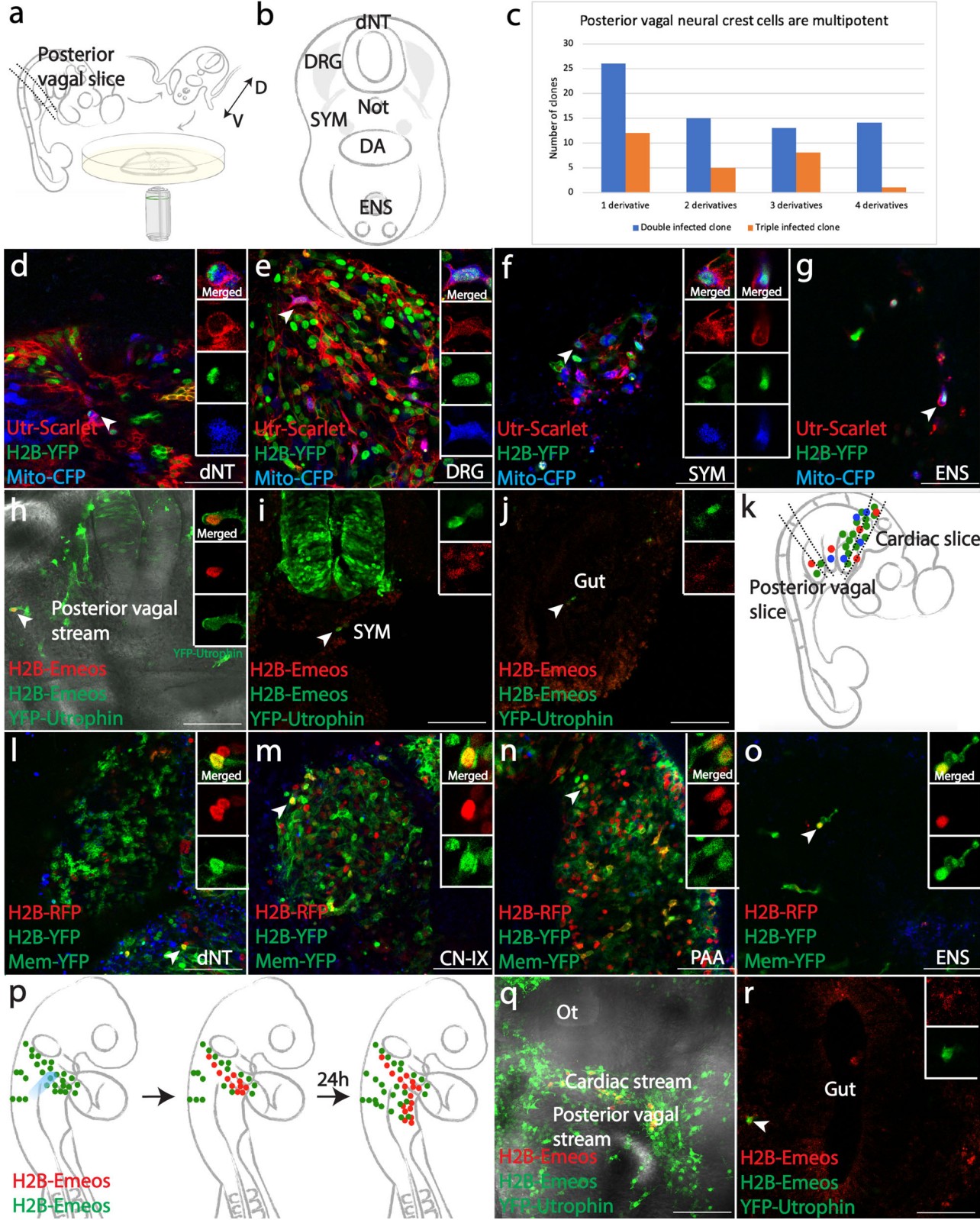

To validate the types of derivatives formed by clonally related cells, we used several markers that reflect mature cell type markers. Antibodies against HuC/D and TuJ1 were used to detect mature neurons, tyrosine hydroxylase (TH) for sympathetic neurons, P0 for Schwann cells and smooth muscle actin (SMA) as a mesenchymal cell fate marker in mature pharyngeal arch arteries. Analysis of immunofluorescence showed that at E7, clonally related cells that localized in different derivatives expressed distinct cell fate markers. For example, in the same embryo, H2B-RFP, H2B-YFP and Mito-CFP infected cells represent a rare clone. Around the CN-IX, cells of this clone gave rise to neuronal (Supplementary Fig. 2a, c), glial (Supplementary Fig. 2b), and Schwann (Supplementary Fig. 2d) cells, as well as in rare cases, sympathetic (Supplementary Fig. 2k)

**Fig. 3 Posterior vagal neural crest cells are multipotent and converge with cardiac neural crest cells to form enteric neurons. a** Schematic diagram of in vivo slice culture for multiplex clonal analysis at the posterior vagal level. At HH21, a ~500 μm slice was obtained at somites 4-7, embedded in 1% agarose and imaged. D, dorsal; V, ventral. **b** Schematic of posterior vagal neural crest-derived tissue (gray), including dorsal neural tube (dNT), dorsal root ganglia (DRG), sympathetic chain (SYM) and enteric nervous system (ENS). **c** Distribution of double (blue) and triple (orange) infected clones among multiple derivatives. **d–g** Utrophin-Scarlet, H2B-YFP, Mito-CFP-coexpressing cells (ID: Supplementary Data 1b-81) were identified in all four locations. **h–j** Multipotency of migratory posterior vagal crest cells was confirmed by photoconverting a cell after delamination (**h**). 24 h post conversion, derivatives of the photoconverted cell were observed in future sympathetic chain (**i**) and around the gut (**j**). **k** In HH21 embryos, two slices were obtained for a transverse view of cardiac and posterior vagal crest derivatives. **l–o** Triply infected cells in the ENS shared clonal signature with other cells in the cardiac slice. A H2B-RFP, H2B-YFP, Mem-YFP-positive clone (ID: Supplementary Data 1c-61) was observed in dNT (**l**), CN-IX (**m**), PAA (**n**), and ENS (**o**). **p** The principle of stream-specific photoconversion assay. UV light was used to specifically convert post-otic cardiac crest cells. If cardiac neural crest cells contribute to the ENS, descendants with red fluorescence should be observed around the gut. **q** In an embryo immediately after conversion, cardiac crest cells were specifically labeled in red. **r** 24 h post-conversion, a converted cell was found near the gut, indicating a cardiac origin of some ENS cells. See Supplementary Fig 2l for absence of converted cell in dorsal posterior vagal crests, ruling out the possibility of nonspecific conversion. Scale bars: **d–g**, **l–o** 100 μm; **h-j**, **q**, **r** 160 μm.

lineages. In the gut, a portion of the cells differentiated into enteric neurons (Supplementary Fig. 2e, g) and Schwann cells (Supplementary Fig. 2h). While clonally related cells differentiated into smooth muscle of the pharyngeal arch arteries, the HuC/D-negative cells in the gut did not form smooth muscle (Supplementary Fig. 2f). Ventrally into the pharyngeal arches and the heart, some cells contributed to smooth muscle of the arteries (Supplementary Fig. 2i) and neurons in the outflow tract (Supplementary Fig. 2j).

As an alternative means of assessing whether cells derived from the cardiac level migrate from the pharyngeal junction posteriorly into the gut, we photoconverted a stream of migrating cardiac crest cells within the arches (Fig. 3p, q). 24 h later, photoconverted progenies were observed in the gut region posterior to the original cardiac crest (Fig. 3r) but absent from dorsal regions of the same embryo where the posterior vagal stream is located in (Supplementary Fig. 2l). This result independently confirms the cardiac crest origin of these photo-labeled enteric cells.

**Cells individually migrating from cardiac and posterior vagal neural crest streams intermix.** To gain insight into the cellular mechanisms underlying multipotency, we optimized experimental conditions of tissue slice cultures[15,19], and performed in vivo time-lapse imaging to capture cellular behaviors. We first focused our analysis on migrating neural crest cells expressing membrane-YFP at both the cardiac (Fig. 4a, Supplementary Fig 3a, Supplementary Movie 1) and posterior vagal levels (Fig. 4g, Supplementary Fig 3b, Supplementary Movie 2). Visual inspection suggested that cells migrated randomly as individuals, though as a population they moved directionally from dorsal to ventral. The heterogeneous movement of the population necessitated quantitative analysis. As such, we conducted squared displacement (SD) analysis, a widely used method to estimate the nature of object motion, which showed that cells exhibited a mixture of directed movement, constrained movement, and free diffusion (Fig. 4b, h). As a consequence, little correlation was observed between the original positions of the cells and their total displacement (Fig. 4c–e, i–k). Sister cells were typically far apart after separation (Fig. 4f, l, m). Based on these findings, we propose that early individual cell migration is stochastic in nature, leading to a random clonal distribution along the dorsoventral axis. However, clonally related cells at both the cardiac and the posterior vagal levels experience disparate environments, which may promote differentiation into distinct cell types at later developmental stages.

If cell migration occurs randomly along the anterior-posterior axis, the cardiac and posterior vagal streams, rather than migrating as separate populations[8,9], are likely to intermix. To test this possibility, we examined cell behaviors at the

pharyngeal junction where cardiac neural crest cells start to migrate into heart and enteric structures. When Utrophin-Scarlet or H2B-YFP expressing cardiac neural crest population was imaged in the lateral view, we noticed that as the stream reached pharyngeal arch 6, some cells began to migrate posteriorly along the developing gut (Fig. 4n, Supplementary Movie 3; n′, Supplementary Movie 4). Taken together, these studies strongly suggest that stochastic cell migration is an important factor and may contribute to the generation of the diverse cell types from cardiac and posterior vagal neural crest cells (Fig. 4o).

**Perturbation of FGF, CXCR4, and RET signaling identifies environmental influences on the directional migration of cardiac neural crest cells.** Finally, using retrovirally mediated molecular perturbations, we explored how cardiac and posterior vagal neural crest cells may differentially respond to environmental cues during early migration and particularly at the pharyngeal junction before entering the heart or the gut. Several signaling pathways have been suggested to play a role in aspects of vagal neural crest migration including FGF, ENDOTHELIN, NT3, EGF, SDF, and GDNF signaling[9,23,24]. To test whether these may differentially affect cardiac versus posterior vagal neural crest populations, we generated recombinant virus overexpressing dominant-negative (DN) mutants of receptors involved in these signaling pathways: FGFR1, CXCR4, RET, EDNRB, TRKC, ERBB2 and tested their roles in vivo. This approach has the advantage of placing identifiable mutant neural crest cells in an otherwise normal environment. Of the DN mutants tested, positive results were obtained for Fgfr1[25,26], Cxcr4[27–29], and Ret[30,31](Fig. 5a), corresponding to signaling mediated by Ffg8, Sdf1 and Gdnf ligands. The results show that vagal neural crest cells respond to these signaling pathways in temporally and axial level-specific manners.

First, we tested the effects of perturbing FGF signaling. Blocking Fgfr1 resulted in little or no migration of mutant cardiac neural crest cells, causing an absence of mesenchymal cells in pharyngeal region at E3 (Fig. 5b, c) and paused development afterwards. Thus, Fgfr1 is required for cardiac neural crest cells during early migration.

In contrast to blocking FGF signaling, cells with dominant negative mutant forms of Cxcr4 and Ret underwent normal initial migration into the pharyngeal arches (Fig. 5d, e for DN-Cxcr4; Fig. 5f, g for DN-Ret). In the long term, however, abrogation of CXCR4 signaling blocked the migration of cardiac neural crest cells into the heart (Fig. 5j–k) but had no effect on migration to or formation of proximal ganglia and pharyngeal arches (Fig. 5h, i) or invasion of the gut (Fig. 5l). As there is variation in virus titer, the numbers of DN-mutant virus infected cells cannot be directly

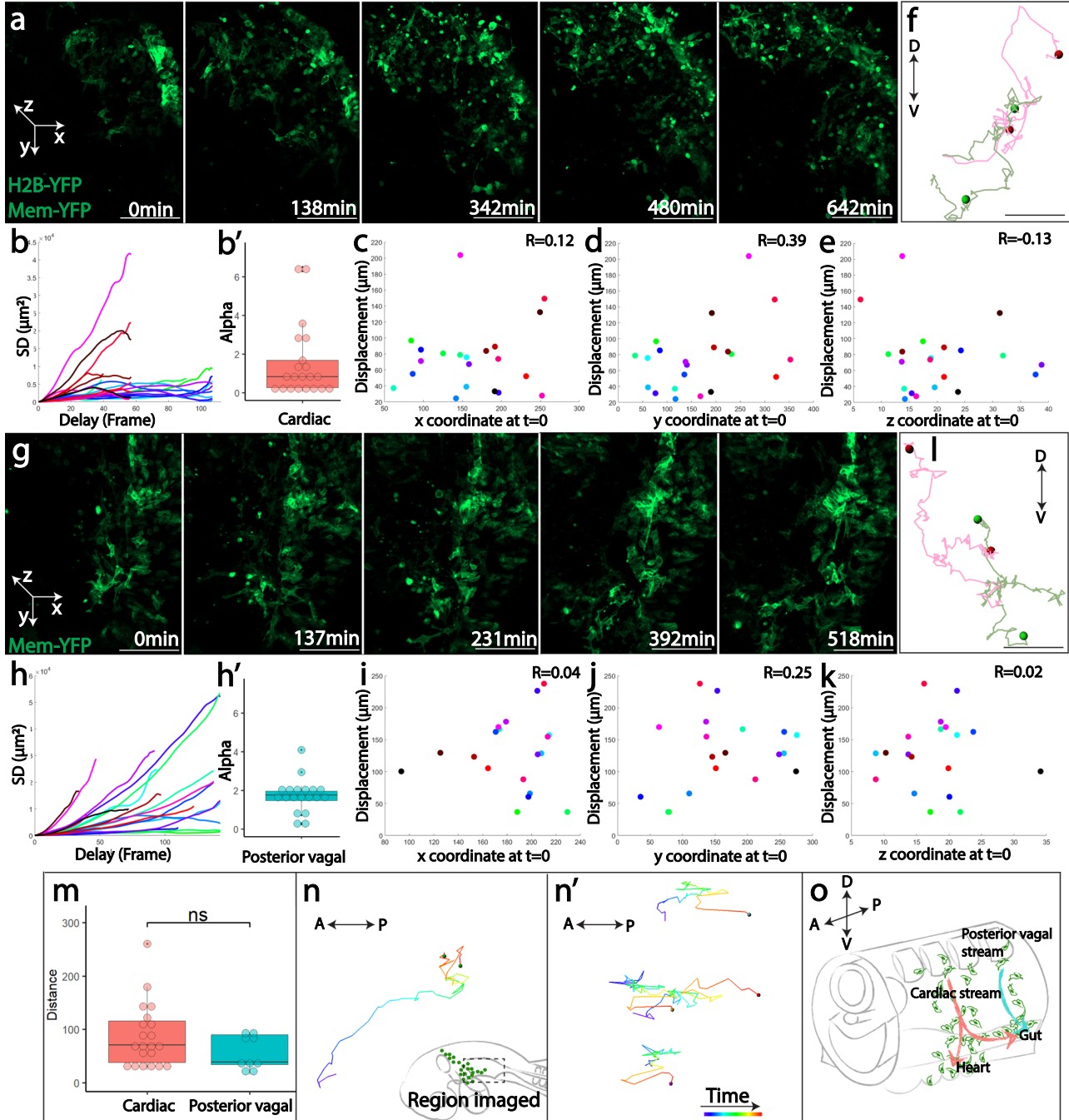

compared with cells infected by control virus. Therefore, the number of cells in the gut was normalized to those in the cranial ganglia to obtain a ratio. At E7, these ratios are similar for *CXCR4*-perturbed cells (13.3%) and control cells (13.9%), suggesting that the effect of *CXCR4* loss-of-function is likely to be specific to cardiovascular migration (Supplementary Data 2). On the other hand, cardiac neural crest cells overexpressing DN-Ret were able to form proximal ganglia and contribute to the cardiovascular system (Fig. 5m–p), but failed to populate the foregut (Fig. 5q).

In order to compare the distance that *RET* mutant cells and normal cells migrated along the gut, we co-injected DN-Ret virus together with H2B-RFP control virus. We first checked infection specificity by examining labeling in the neural tube of the targeted axial levels. Both cardiac specific injection (at somite level 1–3) and vagal injection (at somite level 1–7) showed that control and

*RET* mutant cells reached the anterior foregut proximal to the pharyngeal junction (Fig. 5r, t, An. foregut). However, the majority of cells that invaded the more posterior foregut were control cells (Fig. 5s, u, Foregut). Quantitative analysis shows that DN-Ret infection resulted in an approximately 86% reduction of cells in the foregut, compared with H2B-RFP control (Supplementary Data 2). Interestingly, although both arrived at the pharyngeal arches, cells expressing DN-Ret predominantly formed smooth muscle cells lining the arteries rather than migrating posterior-ventrally into the gut, contrasting with their H2B-RFP expressing counterparts. Furthermore, *RET* dependency is likely to be specific to the cardiac neural crest, because in embryos where the entire vagal neural crest population was infected, the midgut was populated by DN-Ret expressing cells (Fig. 5v), which are derived from posterior vagal neural crest cells[32].

**Fig. 4 Vagal neural crest cells display individual and stochastic early cell migration. a**, **g** Selective snapshots of the migratory cardiac **a** and posterior vagal **g** neural crest cells expressing membrane-YFP. In the coordinate system defined to facilitate quantitative analysis, dorsal-most position in the neural tube was set as the origin. The y axis was defined parallel to the dorsoventral (DV) direction. The mediolateral (ML) axis aligns parallel to the x axis. The z axis runs in the anteroposterior direction (AP). See Supplementary Fig. 3a, b for snapshots with brightfield channel displaying tissue morphology. **b**, **h** Squared displacement (SD) analysis of cell trajectories shows that both cardiac (**b**, $n = 21$ cells examined over 2 time-lapse imaging experiments) and posterior vagal (**h**, $n = 18$ cells examined over 1 time-lapse imaging experiment) neural crest cells exhibit three types of migration, including free diffusion ($\alpha \approx 1$), directed motion ($\alpha > 1$) and constrained motion ($\alpha < 1$). $\alpha$ values are shown in b' and h' for cardiac and posterior vagal populations, respectively. **c–e**, **i–k** The correlation between cell positions in x (**c**, **i**), y (**d**, **j**), z (**e**, **k**) axis at time 0 and their total displacement was plotted for both cardiac (**c–e**) and posterior vagal (**i–k**) neural crest cells. The close-to-0 values of correlation coefficients (R) suggest no notable correlation between initial positions and distances traveled. **f**, **l** Examples of sister cell trajectories at the cardiac (**f**) and posterior vagal (**l**) levels. The dots represent the final positions of sister cells. D, dorsal; V, ventral. **m** The distance between sister cells by the end of imaging were measured. There was no statistically significant difference between the cardiac ($n = 21$ cell divisions examined over 2 time-lapse imaging experiments) and posterior vagal ($n = 9$ cells examined over 1 time-lapse imaging experiment) population (two-sided rank-sum test). (**n**, **n'**) Cell trajectories of cardiac crest migration from pharyngeal arch to the gut (**n**, replicate n') along the anteroposterior axis. The region imaged is shown in the dashed box of (**n**). Color coded timescale (from purple to red) is presented in (n'). A, anterior; P, posterior. **o** Summary of vagal neural crest migration with a convergence of cardiac neural crest and the posterior vagal crest to populate the ENS. In box-and-dot plots (b', h', m), the box bottom, median line, and box top represent the 25th (Q1), 50th (Q2), and 75th percentile (Q3), respectively. Whisker ends represent Q1–1.5*IQR and Q3 + 1.5*IQR, respectively. IQR is interquartile range (Q3-Q1). Scale bars: **a**, **g** 100 µm, **f**, **l** 30 µm.

## Discussion

Our study shows that many caudal hindbrain neural crest cells are multipotent and further demonstrates that individual cardiac neural crest precursors can give rise to both mesenchymal cardiovascular derivatives and neuronal ENS derivatives. Even at early migratory stages, single-cell photoconversion confirms multipotency and shows that daughter cells often take distinct migration routes post-division. This serves as a mechanism to randomly distribute multipotent neural crest cells prior to differentiation. Moreover, dynamic imaging accompanied by stream-specific photoconversion demonstrates that neural crest cells from different migratory streams intermix. Specific signaling cues then guide multipotent cardiac neural crest-derived cells toward specific organs.

At odds with arguments for lineage restriction[33,34], our findings in both the cardiac and posterior vagal axial levels are in good agreement with the idea that neural crest cells are largely multipotent, as well documented for neural crest cells at the trunk level[2,35,36]. While there is apparent heterogeneity within delaminating vagal neural crest population, recent single cell RNA-seq analysis of the vagal neural crest gene regulatory network has identified a $SOX10^{high}/FOXD3^+$ subpopulation that appears to contribute to neural, neuronal and mesenchymal fates[37]. This early SOX10/FOXD3 cluster is likely to represent the multipotent population captured by our clonal analysis. According to our findings, ~20% of individual cardiac crest clones contribute to two or more different neural crest sites as well as the dorsal neural tube, demonstrating broad developmental potential. Although most neural crest cells migrate from dorsal to ventral, there is also movement along the anterior-posterior axis particularly within the pharyngeal arches. As a consequence, our analysis likely reflects an underestimate, as some progeny may not be included in transverse slices used in our analysis. In addition, temporal differences in delamination may account for variation in clone size since cells delaminate from the neural tube for an approximately 24-h period. Cells emerging earliest can migrate the furthest to populate the most distal structures and may undergo more cell divisions than those emerging later. This is consistent with findings using WNT1-CRE and P0-CRE to conduct population level lineage tracing of cardiac neural crest in mice[38], which showed that the cardiac neural crest is comprised of a mixture of differentiated and undifferentiated cells. Here we extend this finding by dissecting the clonal relationship within the mixture of cells. Confetti technology has been used to look at later clonal relationships within enteric ganglia and suggest that there is localized clonal expansion and interaction with neuroectodermal

cells, with some clones contributing to diverse types of neurons and glia[39]. Our study complements this previous work by taking the study to a single cell level at early developmental stages.

Posterior vagal crest clones appear to be tightly related, with 38% of clones found in multiple neural crest sites, compared with 21% of the cardiac crest clones. This may reflect intrinsic differences in cell behavior between these two populations. Posterior vagally-derived tissues are enclosed within a smaller physical compartment, whereas cardiac crest cells are more loosely arranged mesenchymal cells that can easily migrate into adjoining branchial arches. In contrast to the cardiac neural crest, the posterior vagal neural crest does not contribute to cardiovascular derivatives, but exclusively gives rise to sympathetic chain ganglia and ENS cells.

Our clonal analysis shows that progeny of an individual cardiac crest cell not only migrate into the cardiovascular system, but also intermix with the posterior vagal crest and migrate posteriorly to adopt enteric ganglion fate. Amongst all candidate genes that we screened, FGFR1, CXCR4, and RET all influenced the migration of caudal hindbrain neural crest cells, with the latter two apparently required for the decision to migrate toward the heart versus the gut, respectively. Previous studies have shown that FGF signaling is essential for the cardiac neural crest fate decision[23], which is mediated by a low level of Fgf8 secreted at midbrain-to-hindbrain boundary[40–42], as well as during early migration[23], which is affected by high FGF8 expression in the pharyngeal arches[43]. Our results agree with and extend this by showing that blocking Fgfr1 results in abrogation of cardiac neural crest-derived pharyngeal mesenchymal cells. Once the cardiac neural crest cells have entered the pharyngeal arches, further migration into the heart and the gut requires a different subset of signaling molecules. Our results suggest that CXCR4 signaling selectively promotes migration to the heart, consistent with the coordinate expression of CXCR4 by cardiac neural crest cells and its ligand SDF1 in the ectoderm[44]. Indeed, loss of CXCR4/SDF1 signaling has been shown to cause conotruncal defects in mice[45] and chick[28]. Moreover, we find that it does not affect migration into the gut, suggesting a cardiac-specific chemoattraction. At early stages, the CXCR4 and RET mutant phenotype is less penetrant, likely due to delayed generation of mutant protein after viral integration which takes more than 24 h[28]. In contrast to CXCR4, RET is required for cardiac neural crest cells to invade the gut, under the guidance of its ligand GDNF expressed in the gut mesenchyme[46], but has no influence on cells migrating to the cardiac outflow tract. RET deficiency results in reduced number of enteric neurons[47,48]. Since the dominant negative receptors are

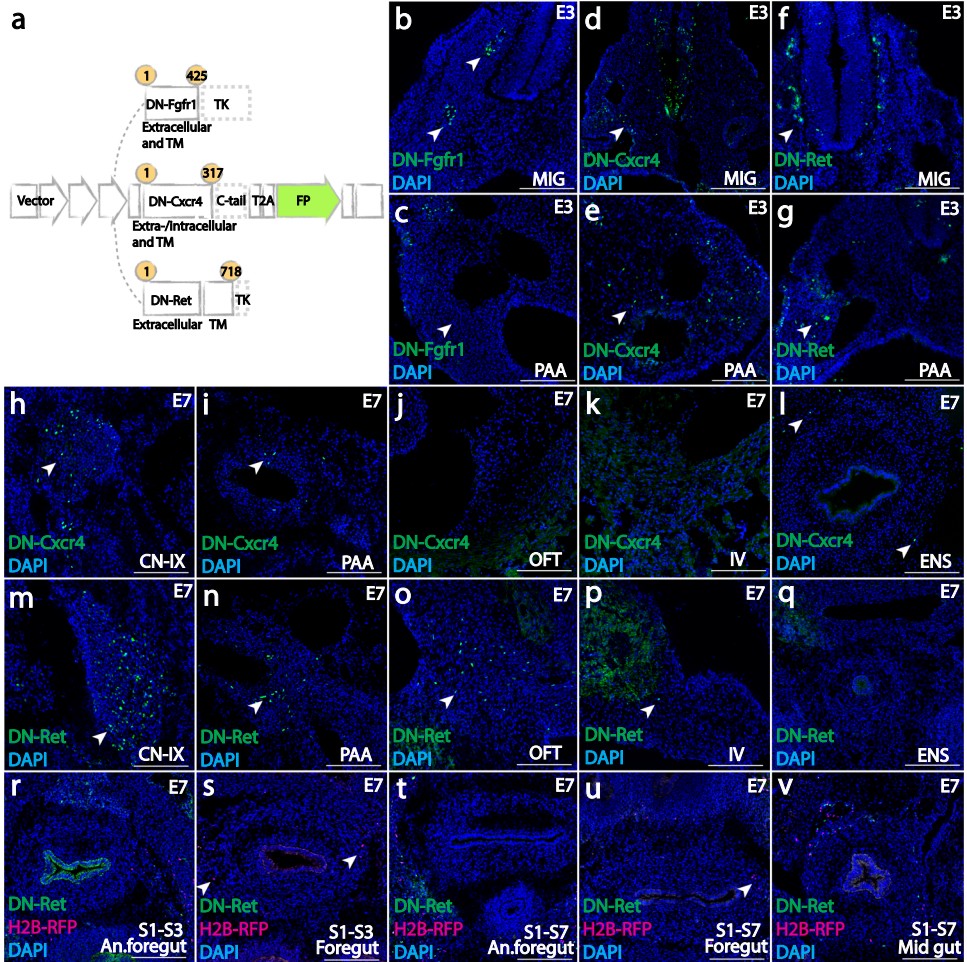

**Fig. 5 Molecular cues required for vagal neural crest cells to adopt cardiac and enteric fates. a** Viral constructs over-expressing dominant-negative mutants of Fgfr1 (containing amino acids (aa) 1–425 of the full-length receptor) and Ret (aa 1–718) by truncating the tyrosine kinase domain, and Cxcr4 (aa 1–317) with c-tail eliminated. The ranges of amino acids are presented in yellow circles. All mutant constructs are labeled with fluorescent protein (FP, green). **b, c** Blocking *FGFR1* signaling restricted cardiac neural crest cells within the ganglia (**b**, 3/3) and prevented pharyngeal mesenchyme formation (**c**, 6/7) at E3. (**d–g**) DN-Cxcr4 and DN-Ret had no apparent effect on initial migration into the ganglia or pharyngeal arches (d (2/2), **e** (5/5) for Cxcr4; **f** (3/3), **g** (8/9) for Ret). **h–l** At E7, DN-Cxcr4 expressing cells formed cranial ganglia (**h**, CN-IX, 2/2), cells around pharyngeal arch arteries (**i**, PAA, 7/7), and enteric nervous system (**l**, ENS, 6/7). However, no DN-Cxcr4 expressing cells gave rise to the outflow septum (j, OFT, 5/5) or interventricular septum (k, IV, 2/2). (m–q) DN-RET expressing cells formed cranial ganglia (**m**, CN-IX, 4/4), cells around pharyngeal arch arteries (**n**, PAA, 6/6), cardiac outflow tract (**o**, OFT, 9/9) and interventricular septum (**p**, IV, 2/2); however, cells failed to enter the foregut (**q**, ENS, 7/9). (**r–v**) The influence of *RET* signaling on vagal neural crest cells is restricted in the cardiac population. Co-injection of DN-Ret and H2B-RFP (magenta) control virus into somite 1-3 showed that despite both mutant and control virus-infected cells were present in anterior foregut (**r**, An. Foregut, 8/8) near the pharyngeal junction, DN-Ret expressing cells failed to enter the foregut (**s** 13/15). When the entire vagal population (S1–S7) was infected, both control and Ret mutant cells reached the anterior foregut (**t**, An. Foregut, 10/12). However, only control cells invaded the foregut (**u**, 8/9). DN-Ret expressing cells arising from posterior vagal crest cells was able to populate the midgut (**v**) (10/10). Numbers in parenthesis shows the number of images with the phenotype presented in the figure out of all images taken. Scale bars: **b–v** 160 µm.

only synthesized at a high level after 1 day, the infected cells are undergoing active migration to the branchial arches prior to onset of the inhibitory effect. Moreover, a similar ratio of dominant-negative and control virus-infected cells are present in cranial nerve nine and the pharyngeal junction, where cell migration is not affected. Therefore, the effects of *CXCR4* and *RET* are more likely to be on their direction of migration rather than survival.

We posit that slight differences in reception of *CXCR4/SDF* versus *RET/GDNF* signaling amongst cardiac crest-derived sister cells within the pharyngeal junction may bias their migration toward either the heart or the gut, thus reflecting the mechanism contributing to their multipotency and diverse derivatives. Initially stochastic cell migration accounts for the disparate distribution of clonally related cells along the dorsoventral axis, resulting

in a random assortment of cells with broad developmental potential within the pharyngeal junction. In response to directional signals, a subset of cells migrates to the heart or the gut and differentiates according to environmental cues therein. Altogether, these results suggest that caudal hindbrain neural crest cells truly represent a multipotent stem cell population with signals from the local environment dictating their final cell fate choice (Fig. 6).

## Methods

**Molecular cloning and virus preparation.** AscI, SpeI, NotI were inserted upstream of ClaI site to create a modified RIA vector. H2B-YFP (#96893), H2B-RFP (#92398), Mito-CFP (#36208), Membrane-YFP (#56558), Utrophin-Scarlet (#26739)[16], H2B-CFP (#25998), Emeos (#73792) encoding plasmids were obtained

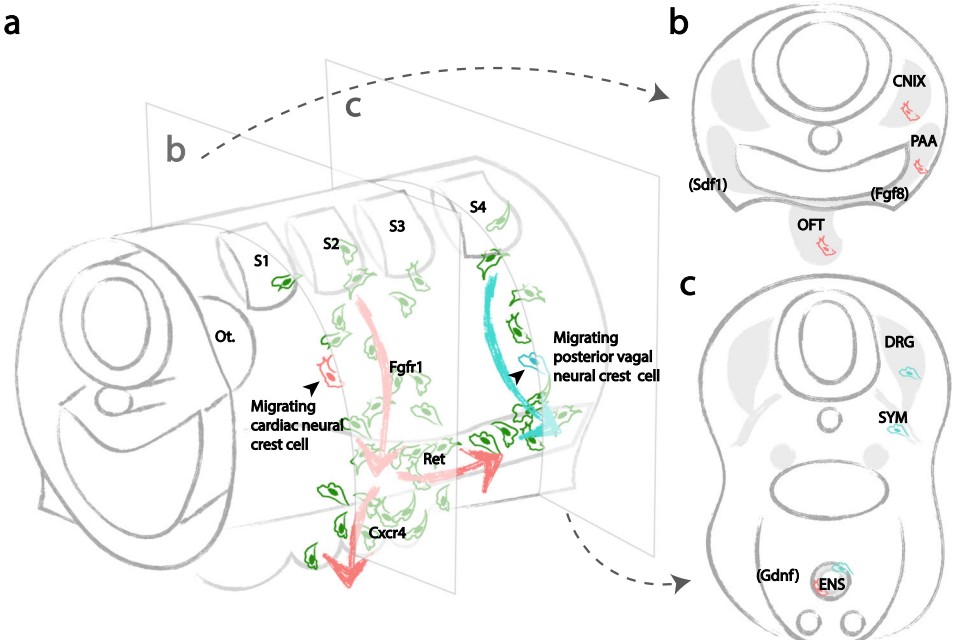

**Fig. 6 Developmental potential of individual vagal neural crest stem cells and bidirectional cardiac neural crest migration driven by molecular cues.**
**a** Early cardiac neural crest cells (red) from the mid-otic to somite 3 exhibit stochastic migration. A subset of randomly selected progeny migrates into the pharyngeal junction in an *FGF*-dependent manner, and subsequently in the direction of the heart or the gut in response to *CXCR4* or *RET*, according to distribution of environmental signals (*FGF8* expressed in pharyngeal arches, *SDF1* expressed by ectoderm in coordination with migrating neural crest, and *GDNF* expressed in gut mesenchyme). In contrast, posterior vagal neural crest cells emigrating from somite 4–7 (blue) are less sensitive to *RET* signaling. **b** As a result, individual premigratory and migratory cardiac neural crest cells are multipotent, capable of giving rise to all cardiac crest-derivatives. **c** Individual premigratory and migratory posterior vagal neural crest cells are multipotent, contributing to diverse components of the sympathetic nervous system. The ENS is composed of neural crest cells of both cardiac and posterior vagal clonal origins.

from Addgene. For retroviral lineage analysis, H2B-YFP, H2B-RFP, Mito-CFP, Membrane-YFP, Utrophin-Scarlet, H2B-CFP were inserted between Not1 and Cla1 sites. To create photoconvertible construct, Emeos protein linked with H2B was inserted in between AscI and SpeI; YFP-Utrophin was inserted between Not1 and Cla1. For molecular perturbations, *DN-FGFR1* (#80431), *CXCR4* (#98967), and *RET* (#23906) plasmids were obtained from Addgene. Truncated *FGFR1* and *CXCR4* receptors were cloned from these templates and subsequently inserted between Not1 and Cla1 sites in RIA. Truncated *RET* was inserted between Asc1 and Spe1 sites in RIA. Primer sequences are displayed in Supplementary Table 1. To produce recombinant RIA virus, plasmids for RIA and ENV-A (a gift from Dr. Connie Cepko) were transfected into chick DF1 cells (ATCC, Manassas, VA; #CRL-12203, Lot number 62712171, Certificate of Analysis with negative myco-plasma testing available at ATCC website) in 15 cm culture dishes. 24 h post-transfection, 12 ml of supernatant containing the virus was collected every 12 h for 4 days, kept at −80 °C until concentration. For each viral concentration, 48 ml of supernatant was centrifuged at 75,500 × *g* for 1.5 h. After centrifugation, 20–30 μl of DMEM was left in the tube to dissolve the pellet containing RIA. The resulting viral stock has a titer of $1 \times 10^7$ pfu/ml. Viral stocks were stored in −80 °C until injection.

**Multiplex retroviral lineage analysis.** To make the working solution, equal volumes of RIA viral stocks encoding H2B-YFP, H2B-RFP, Mito-CFP, H2B-CFP, Membrane-YFP, and Utrophin-Scarlet were mixed. 0.3 μl of 2% food dye (Spectral Colors, Food Blue 002, C.A.S# 3844-45-9) was added into the viral mixture as an indicator. ~0.5 μl of working solution was injected into the lumen of the neural tube adjacent to mid-otic until somite 3 to label cardiac crest, or from mid-otic to somite 7 to label the vagal population. For in vivo clonal analysis (*n* = 35), chick embryos were sealed after injection, incubated at 37 °C for 2 days, harvested at Hamburger Hamilton (HH) Stage 21, screened for specific labeling in the neural crest. A ~500 μm thick transverse slice at cardiac (*n* = 14), posterior vagal (*n* = 12) or both (*n* = 9) regions were cut. Locations with virally labeled cells were imaged using 20x lens on confocal microscope (Zeiss LSM 800). In cardiac clonal analysis, dorsal neural tube, cranial nerve nine, pharyngeal arches, and outflow tract were imaged; in posterior vagal clonal analysis, dorsal neural tube, dorsal root ganglia, sympathetic ganglia, and the gut were imaged. For cardiac clonal analysis of both cardiac and enteric derivatives, cranial nerve nine, pharyngeal arches, and outflow tract from the cardiac slice, as well as the gut from posterior vagal slice were imaged. In those posterior vagal slices, the dorsal regions were not analyzed because the signal was absent (injection specificity screened based on this

criterion). For immunohistochemistry (*n* = 12), embryos were fixed at E7 in 4% PFA in PB for 30 min at 4 °C, embedded in O.C.T compound (Sakura #4583) and cryo-sectioned (*Microm* HM550 cryostat).

**Single-cell and stream-specific photoconversion.** Photoconvertible construct encoding H2B-Emeos and YFP-Utrophin was injected into the lumen of the neural tube adjacent to where the targeting neural crest emerges. At HH13, a transverse slice spanning 2 somites was obtained, immobilized in glass imaging dish. 405 nm laser was used to selectively convert H2B-Emeos from green to red in one migrating cardiac neural crest cell per slice for single-cell photoconversion (*n* = 11), one migrating posterior vagal neural crest cell for single-cell photoconversion (*n* = 5), or all post-otic migrating cardiac crest cells for stream-specific photo-conversion (*n* = 2). Immediately after conversion, an image with z-stack spanning the tissue depth above and below the converted cell was obtained as a proof of specificity. Slice was then placed back to the incubator at 38 degree Celsius. 24 h post incubation, slices were fixed in 4% PFA in PBS for 15 min at 4 °C, embedded in O.C.T compound (Sakura #4583) and sectioned (*Microm* HM550 cryostat) for cell fate analysis. Photoconverted progenies were identified based on a simulta-neous presence of red nuclear and YFP-Utrophin signals. Although blood cells can also produce red nuclear-like autofluorescence, these cells were excluded from the analysis due to lack of YFP-Utrophin signal at the cortical region (Supplementary Fig. 1e, f).

**In vivo time-lapse imaging.** In HH9 embryos, cardiac neural crest from the mid-otic placode to somite 3, or vagal neural crest from mid-otic placode to somite 7 were labeled with ~0.5μl of viruses described above. Embryos were harvested 22 h post injection (HH13), sliced, and cultured in neurobasal media (GIBCO #12349-015). Transverse view was imaged for cell division tracking for migrating cardiac (*n* = 2) and posterior vagal (*n* = 1) neural crest cells; lateral view was imaged for cell migration along the gut (*n* = 2).

**Retrovirally-mediated molecular perturbations.** RIA-H2BEmeos-DNFgfr1 (*n* = 11), H2BEmeos-DNCxcr4 (*n* = 4), or DNRet-H2BEmeos (*n* = 10, 8/10 injected with H2B-RFP control) were supplemented with 0.3μl of 2% food dye (Spectral Colors, Food Blue 002, C.A.S# 3844-45-9) to make the working solution. ~0.5 μl of working solution with a titer of ~$10^6$ particles per mL was injected into the lumen of the neural tube posterior to mid-otic to somite 3 (S1–S3 in brief) to infect cardiac neural crest, or from mid-otic to somite 7 (S1–S7 in brief) to infect vagal neural crest. Embryos were sealed with surgical tape, continues to incubate at 37 °C

until E3 or E7 for analysis. H2BEmeos was used as a cell marker instead of a photoconvertible protein in these experiments.

**Immunohistochemistry.** O.C.T. compound was removed by incubating the frozen section for 3 min in 1xPBS at room temperature. Tissue sections were then permeabilized with 0.3% vol/vol Triton-X100in 1xPBS. Blocking buffer was prepared with 1xPBS, 5% vol/vol normal donkey serum, 0.3% vol/vol Triton-X100. Sections were incubated with primary antibody at 4 °C overnight (primary antibodies used: 1:500 Mouse anti-smooth muscle actin IgG2a, Sigma-Cat#A2547, clone 1A4 monoclonal; 1:500 Mouse anti-HuC/D IgG2b, Invitrogen-Cat#A21271, clone 16A11 monoclonal; 1:500 Mouse anti-TuJ1 IgG2a, Biolegend-Cat# 801201, clone TUJ1 monoclonal; 1:500 Rabbit anti-Tyrosine Hydroxylase (TH), Millipore-Cat# AB152, polyclonal; 1:10 8-IE7 (monoclonal mouse anti-GFAP IgG1), DSHB, 1:10 3H5 (monoclonal mouse anti-HNK1 IgM), DSHB, 1:10 PAX7 (monoclonal mouse anti-Pax7 IgG1), DSHB, 1:10 IE8 (monoclonal mouse anti-P0 IgG1), DSHB. After primary antibody incubation, sections were washed for 3 times (each 15 min) in 1xPBS, and incubated with secondary antibody for 45 min at room temperature (secondary antibodies used: 1:1000 goat anti-mouse IgG2a 633, #A21136 (polyclonal), 1:1000 goat anti-mouse IgG1 647, #A21240 (polyclonal), 1:1000 goat anti-mouse IgG2b 647, #A21146 (polyclonal), 1:1000 goat anti-mouse IgM 647, #A21238 (polyclonal), 1:1000 donkey anti-rabbit 647, #A31573 (polyclonal), Invitrogen Molecular Probes). Immunostained tissue was imaged using a Zeiss AxioImager.M2 with Apotome.2 or Zeiss LSM 800 confocal microscope.

**Statistics and reproducibility.** For multiplexed retroviral clonal analysis, slice cultures were generated in biological replicates of 14, 12, or 9, respectively, at cardiac, posterior vagal, or both regions. Because correct targeting is a prerequisite for slice culture, all 23 cardiac slices were processed as indicated in Fig. 1d (23/23 slices) and the migration pattern analyzed as in Supplementary Fig. 1b. Figures 1e–h, 3d–g, 3l–o are examples of a triply labeled clone with derivatives found in multiple locations at the cardiac, posterior vagal and both levels. Because viral infection of progenitor cells is random, each clone represents an independent event. Therefore, we provide individual data points for each clone observed (Supplementary Data 1) as well as the distribution of clone types among replicates (Figs. 1i and 3c). Because the probably of triple infection are rare, antibody dilutions were first tested on uninfected embryos. Immunohistochemistry was performed in individual embryos (n = 12 replicates) where a triple labeled clone (H2B-YFP, H2B-RFP, Mito-CFP) was observed in cranial nerve nine (CN-IX), the enteric nervous system (ENS), pharyngeal arch arteries (PAA), outflow tract (OFT), and sympathetic chain (SYM) (Fig. 1j–k, Supplementary Fig 2a-k). Note that Fig. 1j–k, Supplementary Fig. 2a–k are not the same clone. A triple labeled clone with broad distribution as shown in Figs. 1e–h, 3d–g, 3l–o, Supplementary Fig. 2a–k is relatively rare (Figs. 1i and 3c). For clonal analysis via photoconversion, we performed single-cell photoconversion with n = 11 replicates on cardiac neural crest, n = 5 replicates on posterior vagal neural crest cell and n = 2 replicates on all post-otic migrating cardiac crest cells for stream-specific photoconversion. In all 11 cardiac single-cell photoconversion experiments, the converted cells were similar to the one in Fig. 2b (represents 11/11). All slices were subsequently processed for antibody staining to obtain Fig. 2c, d, e. Because the behavior of each photoconverted progenitor cell is independent from other cells, we recorded the locations of each clone as a distribution shown in Supplementary Fig 1g. Note that 8/11 clones are located in more than one derivative. Figure 2d represents 7/11 clones, while Fig. 2e represents a rare case (1/11) because the outflow tract tends to have lower numbers of neural crest-derived cells. Supplementary Fig. 1e, f shows converted signal and autofluorescence from blood cells, representative of all 18 experiments. For single posterior vagal neural crest photoconversion, similar results were observed in 5/5 experiments as shown in Fig. 3h; Fig. 3i–j (one of the sister cell after cell division migrated into the gut) occurred in 2/5 experiments. Results for stream-specific photoconversion shown in Fig. 3q–r, Supplementary Fig 2l were observed in 2/2 experiments. Immunohistochemistry for premigratory Pax7+ cells was performed in n = 3 embryos, with all displaying similar patterns as shown in Supplementary Fig. 1c. Figure 4a and g (same as Supplementary Fig. 3a, 3b) include representative images from 5 evenly distributed timepoints over Supplementary Movie 1 and 2 as an overview of cell migration. Molecular perturbation was performed with n = 11 replicates for DN-Fgfr1, n = 4 replicates for DN-Cxcr4 and n = 10 replicates for DN-Ret (8/10 injected with H2B-RFP control). The numbers for representative images in Fig. 5b–v are presented in the figure legend.

**Imaging data analysis.** Pax7+ and DAPI + cells were quantified using Manual Cell Counting in FIJI v1.0 software. Cell numbers after CXCR4 and RET disruption were quantified using Manual Cell Counting with FIJI v1.0 software. For each embryo, two to three randomly selected transverse sections were quantified. To analyze the role of DN-Cxcr4 in cell migration to the enteric nervous system, numbers of labeled cells were counted in images of the foregut and cranial nerve nine. The average cell number in the foregut was normalized to cell number in the cranial nerve, to account for titer variation between experiments. The resulting ratio was compared between DN-Cxcr4 and control virus injected group. For the role of RET signaling during cardiac neural crest migration into the foregut, the numbers of DN-Ret-expressing and control virus-expressing cells in the foregut

were normalized to their counterparts in the pharyngeal junction (anterior to the foregut). Dynamic imaging data was exported from ZEN (blue edition) software and processed using IMARIS 9.1.0 software. In each timeframe, the center of a cell was identified according to the position of its nucleus. Position information was connected to reconstitute cell trajectory, with the x, y, z coordinates in each frame extracted for further analysis. The squared displacement (SD) analysis was conducted using the Mean square displacement analysis of particle trajectories package in MATLAB R2017a. The SD curves were fitted to a power function $f(x) = b*x^\alpha + c$. $\alpha \approx 1$ implies free diffusion, $\alpha < 1$ implies constrained movement, while $\alpha > 1$ implies directed movement. Plots of initial coordinates against displacement distances were generated using MATLAB R2017a. Box-and-dot plots were generated using ggplot2 package in R. Two-sided Wilcoxon rank-sum test was used for statistical comparison between two groups.

**Reporting summary.** Further information on research design is available in the Nature Research Reporting Summary linked to this article.

## Data availability
The authors declare that all data supporting the findings of this study are available within the article and its supplementary information files or from the corresponding author upon reasonable request.

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

## Acknowledgements

We thank Dr. Carlos Lois and the Biological Imaging Facility of the Beckman Institute for sharing equipment. This work is supported by NIHRO1HL14058 to M.E.B.

## Author contributions

W.T. and M.E.B. conceived the project. W.T. and Y.L designed the experiments. W.T. performed the experiments. Y.L. established single-cell photoconversion and provided help with imaging analysis. A.L. performed quantitative analysis. W.T., Y.L., and M. E.B wrote the manuscript with consultation from A.L.

## Competing interests

The authors declare no competing interests.
