## [Peer Review File · Nature Communications]

Reviewers' Comments:

Reviewer #2:

Remarks to the Author:

The authors set out to determine whether neural crest cells are multipotent or predetermined. This is a classic age old question in the neural crest cell field and this particular study focused on vagal neural crest cells and their contributions to cardiac and, or enteric fates. Using a combination of retroviral clonal analysis and single cell photoconversion in combination with live imaging the authors conclude that environmental influences rather than intrinsic information govern cell fate choice of vagal neural crest cells. The experiments overall are generally well done and the single cell resolution and imaging is impressive.

The concepts the authors are investigating are relatively straightforward and the authors conclude from their studies that:

The majority of cardiac neural crest cells are multipotent, able to generate multiple cell types.

Cervical neural crest cells contrast with cardiac crest by only contributing to ENS and sympathetic ganglia but not to the heart.

A large proportion of cardiac crest cells also contribute to the ENS

Cell migration is an important factor in the generation of the diverse cell types arising from cardiac neural crest cells

Although the authors have used some important and novel tools to address their hypotheses, and thus address key developmental questions at high resolution, the bulk of the conclusions confirm previously published observations and it's really surprising to see such an absence of citations to published literature that have previously focused on these topics.

For example, there's no mention of the clonal analyses of cardiac neural crest cells that were performed in avian embryos by Ito and Sieber-Blum (1991) in which the authors concluded "early migratory cardiac neural crest is a heterogeneous population of cells, consisting of pluripotent cells, cells with a partially restricted developmental potential, and cells committed to a particular cell lineage". The type of differentiated cells analysed in that study included pigment cells, smooth muscle cells, connective tissue cells, chondrocytes, and cells in the sensory neuron lineage. In a follow up clonal analysis study of cardiac neural crest cells (1993) the same authors found the presence of pluripotent neural-crest-derived cells, cells with a restricted developmental potential, and apparently committed cells in the posterior visceral arches (ie during their migration).

Furthermore, the pluripotent cells can generate up to four neuronal and non-neuronal phenotypes. Analyses of TrkC mutants which exhibit multiple cardiac malformations revealed three types of cardiac neural crest cells: stem cells that undergo self-renewal and can generate many cell types, cells that are restricted in their developmental potentials, and cells that are committed to the smooth muscle cell lineage (Youn et al 2003). Subsequent studies from the Robbins lab (2006) provided evidence for cardiac neural crest cells retaining multipotent characteristics late in development through contributions to the mature valves and the cardiac conduction system. Hence the multipotentiality of cardiac neural crest cells has been observed at initiation, during migration and throughout embryogenesis and adult life.

The authors' observations regarding cardiac cell migration being an important factor in generating diverse cell types is also not surprising or really novel give the work of Alan Burns' lab which ablated the vagal neural crest cell population adjacent to somites 1-7 and transplanted back single somite lengths of neural tube. This revealed different degrees of contribution of cardiac neural crest cells to the gastrointestinal tract and by doing this at a single somite length level it also revealed intrinsic proliferation differences in axial sub populations of vagal neural crest cells. In

addition, precise GFP labeling via targeted electroporation in chicken embryos has shown that vagal neural crest originating from somite levels 1-4 commence migration towards and contribute to tissues in the heart and the ENS, while those arising from somite levels 4-7 contribute only to the ENS (Kuo and Erickson, 2011; Espinosa-Medina). These focal experiments demonstrated the timing of vagal neural crest pathway choice and biological implications in their contributions to the heart versus gut derivatives. Thus, it's already been well established whether cardiac and posterior vagal neural crest cells can follow dorsolateral and, or ventral pathways in contributing to either the heart or gastrointestinal tract respectively, which directly underpins their range of potential fates.

What's really missing from this study is some indication of the molecular cues that drive these properties in cardiac versus posterior vagal neural crest cells. Given recent publications from the lab focused on cardiac and vagal gene regulatory networks and through single cell sequencing, it's critical to provide molecular insights into the regulation of the process, for the work to be truly novel and more than higher resolution confirmation of already demonstrated principles. In addition, a lot of the authors' conclusions are based on clone location as an indicator of fate. But tissue contribution is different to cell fate and true indication of cell fate requires mature cell type specific markers. The authors did use HuC/D but so many others could have been employed to validate the multipotentiality of lineage traced cells in their final destinations.

Minor Comment

I find the authors' use of the term "cervical" neural crest confusing. Anatomically it doesn't make a lot of sense. The somite 1-7 region is by virtue of its generation of cervical vertebrae, the cervical region, not just somites 4-7. Given the current vernacular of cardiac and vagal neural crest cells it would be more appropriate and consistent for the field to refer to the somite 4-7 region as the posterior vagal neural crest cell population and the somite 1-3 region as cardiac or anterior vagal if you want to make a clear distinction.

Reviewer #3:

Remarks to the Author:

In this article, Tang and colleagues examine the fate of individually labeled neural crest cells emerging from post-otic levels, in chick embryos. They ask whether these progenitors, taken prior to delamination from the neural tube, or during early migration, are multipotent or predetermined towards neuronal or mesenchymal fates. In addition, they examine if the most anterior population also forms enteric neurons.

The first question addresses a partially solved debate in the field, about multipotency or predetermined status of premigratory neural crest cells. While in mouse, elegant works using the Confetti mouse have shown that trunk neural crest cells are multipotent [Baggioni et al], several studies in chick embryos, also in the trunk, argue in favor of either hypothesis. Using a novel retroviral-based stochastic multicolor labeling, the authors recently proposed that trunk neural crest was multipotent in chick as in the mouse [Dev. Biol 2019, doi.org /10.1016/j.ydbio.2019.03.007]. In this previous paper, they also examined briefly the rhombencephalic levels detailed here. Here they extend their 2019 analysis to the post-otic neural crest, where ectomesenchymal derivatives are produced in addition to neuronal fates.

The second part of the study explores whether post-otic neural crest cells anterior to the third somite contribute to enteric nervous system in addition to ectomesenchymal contribution at cardiac levels. As detailed below, this was known at the level of neural crest population (Le Douarin and Teillet 1973). Here, the study identifies individual bipotent ectomesenchymal and ENS progenitors, in vivo, coming back to the first question (multipotency).

As a whole, this study is interesting for developmental biologists as it extends the demonstration

of early neural crest multipotency to cardiac/cervical levels and brings details about the fates of these multipotent progenitors, confirming previous analyses at population level. Several points, however, are incomplete or need to be rephrased as detailed below.

Main points

1- Described as a major finding of the study, the authors insist that until now only the cervical NC was described to participate to the enteric nervous system. This is inaccurate. The original 1973 fate-mapping publication using quail-chick chimeras (quoted by the authors as Ref 7) clearly describes, in 8 embryos, the contribution of NC taken "from the rhombo-mesencephalic constriction to the 1st somite" to the enteric nervous system.

Consequently, the description of the fates of cardiac and cervical NC in the introduction are incomplete, and statements such as "this study ... further reveals a previously unknown shared lineage for mesenchymal cardiac crest derivatives and neuronal ENS derivatives" are misleading. The data presented here are simply in accordance with previous observations and refine previous knowledge to the individual cell level, in vivo, with a novel imaging technology.

2- The labeling technique raises several issues needing additional controls:

Focal injections in the lumen of the neural tube are difficult experiments. Leakage of the viral solution may occur and result in low-level infection of adjacent tissues, either in the neural tube itself, or in adjacent tissues during healing:

- Although histology is not very clear, how do the authors explain that tissues not derived from neural or neural crest cells are seemingly labeled (e.g. around/in the notochord, the endoderm-derived epithelia) in Figure 1D?

- The second part of the analysis relies upon infecting specific adjacent levels (otic-s3 or s4-s7). How is the injection solution constrained in the neural tube lumen precisely located at a given A-P level (e.g. post-otic to s3) and how is diffusion to adjacent areas prevented? The authors need to show examples, taken as soon as possible after injection, to show the exact AP extent of the labeling, and demonstrate the accuracy and reproducibility of the labeling. Here they barely mention that cervical areas were unlabeled at 48h.

3- While the global principle of the multicolor labeling technique is elegant, the statistical significance of the results depends upon the initial number of cells that are infected, which is never estimated. This potentially severely impacts the conclusions of the study, as the number of valid clones may be much lower than described here:

Specifically, previous validation of the technique by the same authors [Ref 10, Dev. Biol 2019, doi.org/10.1016/j.ydbio.2019.03.007] indicates probability for double and triple infection as low, but not very low (about 1% and 0.1% respectively): if the initial number of infected neural crest progenitors is 5000, as estimated previously for trunk neural crest, 50 double and 5 triple identical combinations are to be expected from independent infection of distinct cells (precise numbers are obtained by an easy probability calculation depending upon initial cell numbers). This expected background is close to the number of clones analyzed here (e.g. for cardiac neural crest about 100 double and 10 triple color clones in Fig 1; for cervical NC about 60 double and 25 triple clones in Fig 3).

Moreover, hindbrain neural crest forms in even larger numbers than trunk neural crest, raising further the background numbers. The authors have to relate their results to the total number of infected cells (e.g. in a standard experiment, shortly after injection) to validate their results: It is likely that only triple-labeled clones will be valid considering this control.

4- The angle of the slice used for cervical neural crest analysis seems to exclude the injected part of the neural tube and its proximal areas (perhaps some ganglia). How do the author analyze these regions?

5- Table S1 contains several clones in two locations, one of which is the dorsal neural tube. Do the authors count those clones as multipotent ones? They seem to rather describe dNT-NC shared

origin than NC multipotency and should be removed from analysis.

6- Discussion: " According to our findings which represent an underestimate of developmental potential, one quarter of individual cardiac crest clones contribute to four different sites, demonstrating extensive multipotency. " Once controls and adjustments are done, all estimates of the degree of multipotency will need to be re-evaluated.

7- In Discussion, third paragraph, what do the authors qualify a "high degree of multipotency"? Three or more NC derivatives, excluding the neural tube? In this case, lower numbers seem to apply from Table S1.

Minor points:

- When describing a clone, indicate clone ID in the text to facilitate link to Table S1.
- Cardiac clone size ranges from 3 cells to 30 cells: this does not seem to be a "uniform"size.

Check spelling:

- Results Part1: "probably" for probability
- Table S1 header: "clone size"

Reviewer #2

General Comments;

The experiments overall are generally well done and the single cell resolution and imaging is impressive. The concepts the authors are investigating are relatively straightforward and the authors conclude from their studies that: The majority of cardiac neural crest cells are multipotent, able to generate multiple cell types. Cervical neural crest cells contrast with cardiac crest by only contributing to ENS and sympathetic ganglia but not to the heart. A large proportion of cardiac crest cells also contribute to the ENS. Cell migration is an important factor in the generation of the diverse cell types arising from cardiac neural crest cells.

We thank the reviewer for his/her positive comments on the manuscript.

Specific Comments

1. *Although the authors have used some important and novel tools to address their hypotheses, and thus address key developmental questions at high resolution, the bulk of the conclusions confirm previously published observations and it's really surprising to see such an absence of citations to published literature that have previously focused on these topics. For example, there's no mention of the clonal analyses of cardiac neural crest cells that were performed in avian embryos by Ito and Sieber-Blum (1991) in which the authors concluded "early migratory cardiac neural crest is a heterogeneous population of cells, consisting of pluripotent cells, cells with a partially restricted developmental potential, and cells committed to a particular cell lineage". The type of differentiated cells analyzed in that study included pigment cells, smooth muscle cells, connective tissue cells, chondrocytes, and cells in the sensory neuron lineage. In a follow-up clonal analysis study of cardiac neural crest cells (1993) the same authors found the presence of pluripotent neural-crest-derived cells, cells with a restricted developmental potential, and apparently committed cells in the posterior visceral arches (ie during their migration). Furthermore, the pluripotent cells can generate up to four neuronal and non-neuronal phenotypes. Analyses of TrkC mutants which exhibit multiple cardiac malformations revealed three types of cardiac neural crest cells: stem cells that undergo self-renewal and can generate many cell types, cells that are restricted in their developmental potentials, and cells that are committed to the smooth muscle cell lineage (Youn et al 2003). Subsequent studies from the Robbins lab (2006) provided evidence for cardiac neural crest cells retaining multipotent characteristics late in development through contributions to the mature valves and the cardiac conduction system. Hence the multipotentiality of cardiac neural crest cells has been observed at initiation, during migration and throughout embryogenesis and adult life.*

We thank the reviewer for providing these references and apologize for not including citations to these important previous studies. We agree that *in vitro* clonal analysis has valuable implications for our work. We note that our study builds upon these previous papers which examined the developmental potential of cardiac neural crest cells explanted from the embryo; by contrast, we explore the fate of individual cardiac neural crest cells in their endogenous environment.

In the revised manuscript, we discuss the *in vitro* clonal analysis of migratory (Ito and Sieber-Blum, 1991) and post-migratory (Ito and Sieber-Blum, 1993) cardiac neural crest cells which showed that neural crest cells are capable of generating multiple cell fates such as pigment cells, smooth muscle cells, connective tissue cells, chondrocytes and sensory neuron. In addition, the potential of cardiac neural crest cells is likely to be influenced by TrkC (Youn et al, 2003). To put these findings in their appropriate context, in our revised paper (page 3-4), instead of stating "little is known about the developmental potential of clonally related neural crest cells at single cell level", we first introduce these insightful results from explant cultures and discuss how these results have led to our clonal analysis, with the aim of extending the analysis *in vivo* to the progeny of individual neural crest cells.

We also refer to the Nakamura et al, 2006, study which utilized Wnt1-Cre and P0-Cre to conduct lineage tracing on cardiac neural crest at a population level. These authors proposed an important concept: "although we have no direct experimental evidence that supports multipotentiality of the labeled NCCs, we speculate that these cells are not fully differentiated at this stage, as evidenced by the lack of mature markers, and may represent cells somewhere along the developmental pathway". The results showing a

mixture of differentiated and undifferentiated cells lead to the question whether the cardiac neural crest is a heterogeneous population possessing diverse developmental potential, which our paper further addresses. In the revised version, we cite this paper in the discussion (page 15 line 21) to emphasize how these open questions can be examined with RIA clonal analysis, and the way our results interact with and enrich the literature.

2. The authors' observations regarding cardiac cell migration being an important factor in generating diverse cell types is also not surprising or really novel give the work of Alan Burns' lab which ablated the vagal neural crest cell population adjacent to somites 1-7 and transplanted back single somite lengths of neural tube. This revealed different degrees of contribution of cardiac neural crest cells to the gastrointestinal tract and by doing this at a single somite length level it also revealed intrinsic proliferation differences in axial sub populations of vagal neural crest cells. In addition, precise GFP labeling via targeted electroporation in chicken embryos has shown that vagal neural crest originating from somite levels 1-4 commence migration towards and contribute to tissues in the heart and the ENS, while those arising from somite levels 4-7 contribute only to the ENS (Kuo and Erickson, 2011; Espinosa-Medina). These focal experiments demonstrated the timing of vagal neural crest pathway choice and biological implications in their contributions to the heart versus gut derivatives. Thus, it's already been well established whether cardiac and posterior vagal neural crest cells can follow dorsolateral and, or ventral pathways in contributing to either the heart or gastrointestinal tract respectively, which directly underpins their range of potential fates.

We agree with the reviewer that previous studies have demonstrated a contribution of vagal neural crest to the heart and gut. In the revised manuscript, we have added a reference to the paper from Burn's and colleagues, as it illustrates distinct developmental potentials along the anterior-posterior axis within the "vagal neural crest" population (page 3 line 22-25). We have also cite Kuo and Erickson, 2011 and Espinosa-Medina et al, 2017 as important background for this study.

The question the reviewer raised made us realize we failed to explain the purpose of this study clearly. These two papers showed that neural crest cells arising adjacent to somite levels 1-3 that give rise to heart and enteric nervous system are two distinct populations with temporally separated migration pathways (Kuo and Erickson, 2011), or Schwann cell precursors migrating along the vagus nerve (Espinosa-Medina et al, 2017). The authors concluded that cardiac and enteric neural crest cells from somite level 1-3 are fate restricted populations. However, focal electroporations and grafts label a small cell population rather than single cells, thus limiting their ability to address whether these cells are from the same individual progenitor before emigration. We address this question with clonal resolution and broad timescale, in the way *confetti* does in genetic systems.

We have revised the introduction to discuss all of the previous papers mentioned in points 1-2. We explain the relationship between these papers and ours, with an emphasis on the new knowledge our paper contributes to. We thank the reviewer for raising this point which has greatly improved the paper and apologize for our previous lack of clarity.

3. What's really missing from this study is some indication of the molecular cues that drive these properties in cardiac versus posterior vagal neural crest cells. Given recent publications from the lab focused on cardiac and vagal gene regulatory networks and through single cell sequencing, its critical to provide molecular insights into the regulation of the process, for the work to be truly novel and more than higher resolution confirmation of already demonstrated principles. In addition, a lot of the authors' conclusions are based on clone location as an indicator of fate. But tissue contribution is different to cell fate and true indication of cell fate requires mature cell type specific markers. The authors did use HuC/D but so many others could have been employed to validate the multipotentiality of lineage traced cells in their final destinations.

The reviewer raises an important point that molecular insights into drivers of cardiac/vagal neural crest migration and cell fate are important. As suggested, we now further validate multipotency by using additional markers that reflect more mature cell type markers. HuC/D is a broadly expressed neuronal marker, so we have now added TuJ1 as a neurofilament marker, tyrosine hydroxylase (TH) for sympathetic neurons, and P0 for Schwann cells in cranial ganglia, cardiac ganglia and the ENS. We have also included

smooth muscle actin as a mesenchymal cell fate marker in mature pharyngeal arch arteries (page 11, line 8-19, FigS2).

We have also tested potential guidance cues that may discriminate between guiding neural crest cells to cardiac versus enteric sites of localization. To this end, we have tested the effects of several candidate cell surface receptors expressed in the vagal neural crest (as identified in various RNA-seq screens). We adapted our RIA virus technology to generate dominant-negative (DN) mutant versions of receptors that encode FGFR1, CXCR4, and RET into our RIA viruses in order to interfere with FGF, SDF and GDNF signaling. Importantly, we find that vagal neural crest cells respond to these signaling pathways in temporal- and axial level-dependent manner. Blocking FGFR1 resulted in restricted or no migration of cardiac neural crest cells at HH14, and an absence of migration into pharyngeal region at E3. Thus, cardiac neural crest cells respond to FGF signaling likely through FGF8 expressed in the pharyngeal arches during early migration. In contrast, initial migration into the pharyngeal arches of neural crest cells mutant for CXCR4 and RET appeared normal. However, abrogation of CXCR4 signaling inhibits neural crest migration into cardiovascular system but has no effect on their invasion of the gut; in contrast, DN-RET expressing vagal neural crest cells from somite levels 1-3 could still enter the outflow tract, but failed to populate the foregut. We thank the reviewer for this suggestion which helped identify environmental cues mediating cell fate decisions. Coupled with our clonal analysis these results suggest that individual neural crest cells within the branchial arches are equally capable of contributing to the heart or gut and that the cell fate decision may be stochastic. Environmental cues influence cell movement to heart (via CXCR4/SDF) versus gut (via RET/GDNF) and in the absence of recognition of one signal, cells are diverted to the other derivative.

Minor Comment

1. I find the authors' use of the term "cervical" neural crest confusing. Anatomically it doesn't make a lot of sense. The somite 1-7 region is by virtue of its generation of cervical vertebrae, the cervical region, not just somites 4-7. Given the current vernacular of cardiac and vagal neural crest cells it would be more appropriate and consistent for the field to refer to the somite 4-7 region as the posterior vagal neural crest cell population and the somite 1-3 region as cardiac or anterior vagal if you want to make a clear distinction.

We apologize for the confusion and thank the reviewer for pointing out this poor verbiage. Sometimes the literature refers to the subpopulation of vagal neural crest from somite 4-7 as "cervical neural crest", which might not be appropriate in this context.

In our revised manuscript, we refer to post-otic, somite 1-3 as "cardiac neural crest", somite 4-7 as "posterior vagal neural crest", and somite 1-7 as "vagal neural crest" or "caudal hindbrain neural crest". We now specify the definition of these terms in the introduction (page 3 line 13-20).

--

Reviewer #3

General Comments:

The first question addresses a partially solved debate in the field, about multipotency or predetermined status of premigratory neural crest cells... Here they extend their 2019 analysis to the post-otic neural crest, where ectomesenchymal derivatives are produced in addition to neuronal fates. The second part of the study explores whether post-otic neural crest cells anterior to the third somite contribute to enteric nervous system in addition to ectomesenchymal contribution at cardiac levels. As detailed below, this was known at the level of neural crest population (Le Douarin and Teillet 1973). Here, the study identifies individual bipotent ectomesenchymal and ENS progenitors, in vivo, coming back to the first question (multipotency). As a whole, this study is interesting for developmental biologists as it extends the demonstration of early neural crest multipotency to cardiac/cervical levels and brings details about the fates of these multipotent progenitors, confirming previous analyses at population level. Several points, however, are incomplete or need to be rephrased as detailed below.

We thank the reviewer for the positive comments.

Specific Comments

1. Described as a major finding of the study, the authors insist that until now only the cervical NC was described to participate to the enteric nervous system. This is inaccurate. The original 1973 fate-mapping publication using quail-chick chimeras (quoted by the authors as Ref 7) clearly describes, in 8 embryos, the contribution of NC taken "from the rhombo-mesencephalic constriction to the 1st somite" to the enteric nervous system. Consequently, the description of the fates of cardiac and cervical NC in the introduction are incomplete, and statements such as "this study ... further reveals a previously unknown shared lineage for mesenchymal cardiac crest derivatives and neuronal ENS derivatives" are misleading. The data presented here are simply in accordance with previous observations and refine previous knowledge to the individual cell level, *in vivo*, with a novel imaging technology.

We thank the reviewer for this comment that made us realize that we were not sufficiently clear in our discussion of previous literature. We agree with the reviewer that previous papers such as 1973 fate-mapping, Kuo and Erickson, 2011, Espinosa-Medina et al, 2017 showed that neural crest cells arising from somite 1-3 level can migrate toward the gut and give rise to the enteric nervous system. The purpose of our paper was to extend these findings (which are on the population level) to analyze the potential of individual neural crest cells.

Kuo and Erickson, 2011 and Espinosa-Medina et al, 2017 suggested that neural crest population from somite levels 1-3 that give rise to heart and enteric nervous system are two distinct thus developmentally restricted populations, following temporally separated migration pathways (Kuo and Erickson, 2011), or a derived from a pool of Schwann cell precursors migrating along the vagus nerve (Espinosa-Medina et al, 2017). However, focal electroporations and grafts label a small cell population rather than single cells, thus limiting their ability to address whether neural crest cells that migrate into the heart and the gut are from the same progenitors before emigration. In the present paper, we sought to explore the developmental potential of single neural crest progenitor and examined their fate over a broader time scale (pre-migratory to E7) using clonal resolution *in vivo*, similar to the way *confetti* does in genetic systems. We also applied photolabeling of a single migrating cell, as a complementary approach to anterograde clonal analysis.

We have revised the introduction to discuss the results from previous studies and explain how conclusion our paper enriches the current understanding of clonal potentiality and fate choice in vagal neural crest cells. We apologize for the confusion and thank the reviewer for this insightful input.

2. The labeling technique raises several issues needing additional controls:
a. Focal injections in the lumen of the neural tube are difficult experiments. Leakage of the viral solution may occur and result in low-level infection of adjacent tissues, either in the neural tube itself, or in adjacent tissues during healing: Although histology is not very clear, how do the authors explain that tissues not derived from neural or neural crest cells are seemingly labeled (e.g. around/in the notochord, the endoderm-derived epithelia) in Figure 1D?

We thank the reviewer for this valuable comment, as focal injection is essential for clonal analysis. It is correct that the neural tube itself is infected. The viral mix fills the lumen, labeling the neural tube including pre-migratory neural crest cells. However, neural crest cells are the only cells that emigrate into the periphery. We agree that injection into non-neuroectodermal tissue would confound the results. Therefore, we screened every embryo to make sure no other tissue (e.g. notochord/mesoderm) were infected. Infected mesoderm is easy to detect since viral signals in compacted mesodermal tissue are much brighter than in neural crest cells. Such unsuccessfully injected embryos were excluded from clonal analysis (also see Methods: multiplex retroviral lineage analysis).

Figure 1D is a collective image from 16 tiles and 10 z-stacks to show the entire explant slice; thus, the magnification is not optimal to distinguish cellular structures. Explant slices tend to autofluoresce in the red channel near the surface. What the reviewer described as signal near the notochord is actually autofluorescence accumulated from multiple z-stacks (red in the notochord and pharyngeal endoderm). In the new Figure 1D, we removed the top and bottom stacks which have most such noise and outlined the notochord. The other signals around the notochord are from Schwann cells near the ventral neural tube (arrow). To clarify these points, we have also amended the figure legend for Fig1D.

b. The second part of the analysis relies upon infecting specific adjacent levels (otic-s3 or s4-s7). How is the injection solution constrained in the neural tube lumen precisely located at a given A-P level (e.g. post-otic to s3) and how is diffusion to adjacent areas prevented? The authors need to show examples, taken as soon as possible after injection, to show the exact AP extent of the labeling, and demonstrate the accuracy and reproducibility of the labeling. Here they barely mention that cervical areas were unlabeled at 48h.

In our experiments, axial specificity was achieved by small injection from the posterior to anterior direction. Thus, if there were leakage to other axial levels, it would be toward the anterior into the cranial neural crest. To label the cardiac crest population, we exclusively injected into the neural tube from somite 3 upward. To label the entire vagal population, we injected from somite 7. Also, we used a small injection volume (~0.5 μ l with food color) to limit the extent of diffusion. Importantly, the virus is only active for limited amount of time (~2 hours); thus, infection is restricted to a particular axial region. During tissue processing, we determined the extent of infection by documenting the signal in the neural tube and excluded embryos with inappropriate infections. According to the reviewer's suggestion, we have incorporated these explanations into the Methods section, and included a whole-mount live embryo imaged immediately after injection at the vagal level (FigS1A), and 48 hours post infection at the cardiac level (FigS1B,B') to demonstrate specificity. We thank the reviewer for this suggestion which helped us demonstrate the accuracy and reproducibility of our injections.

3. While the global principle of the multicolor labeling technique is elegant, the statistical significance of the results depends upon the initial number of cells that are infected, which is never estimated. This potentially severely impacts the conclusions of the study, as the number of valid clones may be much lower than described here: Specifically, previous validation of the technique by the same authors [Ref 10, Dev. Biol 2019, doi.org/10.1016/j.ydbio.2019.03.007] indicates probability for double and triple infection as low, but not very low (about 1% and 0.1% respectively): if the initial number of infected neural crest progenitors is 5000, as estimated previously for trunk neural crest, 50 double and 5 triple identical combinations are to be expected from independent infection of distinct cells (precise numbers are obtained by an easy probability calculation depending upon initial cell numbers). This expected background is close to the number of clones analyzed here (e.g. for cardiac neural crest about 100 double and 10 triple color clones in Fig 1; for cervical NC about 60 double and 25 triple clones in Fig 3). Moreover, hindbrain neural crest forms in even larger numbers than trunk neural crest, raising further the background numbers. The authors have to relate their results to the total number of infected cells (e.g. in a standard experiment, shortly after injection) to validate their results: It is likely that only triple-labeled clones will be valid considering this control.

The reviewer raises an excellent point here. To validate the result, we quantified the number of neural crest precursors in the hindbrain using Pax7, a marker for dorsal neural tube cells with the potential to form neural crest cells. In our revised manuscript, we include a paragraph describing the probability calculation of multiple infections $P\{n\}$ at the vagal axial level (page 6 line 12-page 7 line 9), thus repeating this calculation previously done at the trunk level. This statistical analysis is also described in FigureS1C, D, where $P\{2\} = 0.019$ and $P\{3\} = 0.0014$. According to Figure 4 in our 2019 trunk clonal analysis paper, it is the ratio between Pax7+ cells and total cells in the neural tube that determines the probability of co-infection (~0.25 for hindbrain). Although the numbers of neural crest and neural tube cells are larger in the vagal (hindbrain) than in trunk (spinal cord) levels, their ratio, which determines m and probability of multiple infection, was found to be similar. According to the new statistical evaluation, both double and triple infections can be counted as rare clones for our vagal infections.

4. The angle of the slice used for cervical neural crest analysis seems to exclude the injected part of the neural tube and its proximal areas (perhaps some ganglia). How do the author analyze these regions?

We apologize for the confusion and now clarify what we did and how we cut slices to analyze the regions. Three types of clonal analysis were conducted: first, for cardiac clonal analysis, we injected between mid-otic level to somite 3 and cut slices through the neural tube adjacent to somites 1-3, the branchial arches and the heart. Second, for "posterior vagal" analysis, we injected the virus from mid-otic

level to somite 7 and cut slices through the neural tube between somite 4-7 and the adjoining posterior branchial arches and gut (see Figure 3D-F). Here, we found many double or triple labelled cells in the ENS without clonally related cells in more proximal derivatives on the same slice through the posterior vagal neural tube. Thus, we wondered whether their sister cells might be in cardiac derivatives. Therefore, in the third clonal analysis, we injected the virus into mid-otic level to somite 3, but cut slices at both cardiac and posterior vagal level (Fig3K). Because the virus only infected cardiac crest, there was signal in the neural tube of the hindbrain adjacent to somites 1-3 but no signal in the neural tube at the level of somite 4-7. We also screened for signal in derivatives closer to the neural tube but detected none. We found that the ENS was the only region in posterior vagal slices with labeled neural crest cells (Fig3K). We thank the reviewer for pointing out that we hadn't explained this properly. In our revised methods (multiplex retroviral lineage analysis), we explained the choices of imaging areas more comprehensively.

5. *Table S1 contains several clones in two locations, one of which is the dorsal neural tube. Do the authors count those clones as multipotent ones? They seem to rather describe dNT-NC shared origin than NC multipotency and should be removed from analysis.*

We agree with the reviewer that this type of clone alone cannot be interpreted as indicating neural crest multipotency. Therefore, we counted clones localized in 2 or more neural crest derivatives other than dNT as "multipotent". We recorded the dNT as one site to be consistent with previous papers on the same topic (Baggiolini et al, 2015 for *Wnt1-cre*, Bronner-Fraser and Fraser, 1988, 1989). In this revised version, we have clarified these points in the text (page 7 line 15-16, page 15 line 13).

6. *Discussion: "According to our findings which represent an underestimate of developmental potential, one quarter of individual cardiac crest clones contribute to four different sites, demonstrating extensive multipotency. " Once controls and adjustments are done, all estimates of the degree of multipotency will need to be re-evaluated.*

We thank the reviewer for this comment. We have revised the discussion based on the new statistical analysis and other insights the reviewer provided above.

7. *In Discussion, third paragraph, what do the authors qualify a "high degree of multipotency"? Three or more NC derivatives, excluding the neural tube? In this case, lower numbers seem to apply from Table S1.*

This is a good question that has led us re-evaluate our definitions and conclusions about multipotency. We found the term "high degree" (which was introduced by Baggiolini et al, 2015) inappropriate in the context of this paper, as we were not comparing between the clones. Therefore, we have adjusted the discussion based on our analysis. Instead of using "degree of multipotency", we defined "multipotency" as clones that give rise to at least two derivatives other than the neural tube and revised the discussion accordingly (page 15 line 12-14). We hope this better defines our clonal characteristics and thank the reviewer for these helpful suggestions.

Minor points:

- *When describing a clone, indicate clone ID in the text to facilitate link to Table S1.*

Thank you for this valuable suggestion. We have incorporated clone ID in the figure legends.

- *Cardiac clone size ranges from 3 cells to 30 cells: this does not seem to be a "uniform" size.*

We apologize for the incorrect wording here. We have revised this description as "neural crest-derived clones display variable sizes, with an average of 9 cells per clone, indicating diverse proliferative properties." (page 7 line 14-15). We thank the reviewer for catching this point.

Check spelling:

- *Results Part1: "probably" for probability*

- *Table S1 header: "clone size"*

We thank the reviewer for pointing out these issues and have corrected the spelling in the revised manuscript.

Reviewers' Comments:

Reviewer #3:

Remarks to the Author:

In this revised version of their work, the authors have addressed most of my previous concerns. The photoconversion of individual cells in vivo is particularly impressive, and confirms the results from multicolor labelling nicely. They also have added experimental manipulation of signaling involved in neural crest fate decisions, which add an important aspect to the study. Altogether, while those conclusions were previously deduced from studies on cell populations, their work brings novel data at the single cell scale on the formation of vagal neural crest, of great interest for developmental biologists.

Minor

Recent study by Ling and Sauka-Spengler explored multipotency of vagal neural crest in chick embryos as well, using epigenetic analyses. This study and its conclusions should be put forward in the discussion.

Reviewer #4:

Remarks to the Author:

This manuscript study the old and rather controversial problem of whether neural crest are truly multipotent cells. Using retrovirally mediated multiplex clonal analysis and photoconversion and focusing in the caudal hindbrain neural crest, they conclude that cardiac and vagal neural crest are multipotent. They also explore the signals that could control their migration into specific regions of the embryo, founding that SDF1/CXCR4 is involved in guiding neural crest into the heart, while GDNF/RET seems to guide them into the gut.

This is a nicely executed work that address a very important issue related to cell differentiation. The result are clear and the conclusions sounds.

I am satisfied with the answer given to Reviewer 2 comments.

An important aspects of this new submission is the manipulation of CXCR4 and RET signalling during neural crest migration. However a better quantification of the results need to be included. This will allow to answer important pending questions:
Is the inhibition of CXCR4/RET affecting neural crest migration, survival or both?
Is the proportion of cardiac neural crest that contribute to the gut affected when CXCR4 is inhibited?

Reviewer #5:

Remarks to the Author:

I was asked to comment on points raised by Referee #2, and how the authors have addressed them.

Referee #2 pointed out the primary weaknesses of the paper: Briefly, 1) The work is not truly novel. The authors have used novel tools to confirm at higher resolution (single cell) the multipotentiality of cardiac neural crest cells and their migration, first shown by clonal analysis and somite transplantation from earlier studies highlighted by the referee but not cited; 2) There is little novel insight provided, regarding the nature of molecular cues that drive cell fate, or that could be used to validate lineage tracing that was used to indicate multipotentiality of neural crest emigrating from specific locations adjacent to somites 1-7.

I do not think author's response to the first criticism has strengthened the paper. They have provided the missing citations, as suggested by the reviewer, and used these to focus the rationale and provide context for the study. However the primary criticism remains, that it was already known that neural crest cells derived from the branchial arch region contain a mix of pluripotential neural crest-derived cells, as well as cells that are restricted in their fate, and cells that are committed, and that the pluripotential cells may generate both neurons and non-neuronal cells. This new study drills down on these earlier observations by using novel lineage tracing and imaging tools that enable identification of contributions at the single cell level. Nevertheless, this is a minor advance in what is known about contributions to cardiac and enteric neural crest fates.

The authors provided an adequate response to the second criticism by the addition to the manuscript of neural crest cell fate markers specific to sympathetic neurons and Schwann cells, and muscle actin as an indicator of mesenchymal cell fate. Further, the authors addressed molecular guidance cues specific to localization of cardiac and enteric neural crest. Their analysis suggests that individual neural crest cells coming from the branchial arch region may be equally capable of contributing to either heart or gut. This is a reasonable interpretation and strengthens the study.

Reviewer 3

1. *Recent study by Ling and Sauka-Spengler explored multipotency of vagal neural crest in chick embryos as well, using epigenetic analyses. This study and its conclusions should be put forward in the discussion.*

Thank you for bringing this omission to our attention, for which we apologize. We are well aware of this lovely paper and now have included the appropriate reference in the second paragraph of discussion (page 15 line 11-15).

Reviewer 4

1. *An important aspect of this new submission is the manipulation of CXCR4 and RET signalling during neural crest migration. However a better quantification of the results need to be included.*

The reviewer raises an excellent point. Accordingly, we have added quantification in which we compared the number of neural crest cells in the gut after manipulating CXCR4 signaling and after control virus infection. For DN-CXCR4 and control group, we normalized number of cells in the gut to that in proximal derivative cranial nerve nine, to account for potential titer variation (page 14 line 2-7). Please see question 3 for the result.

We have also quantitatively analyzed the effect of RET signaling on cell migration along the gut by comparing number of cells within the foregut after RET perturbation and control virus infection. To account for variation, we normalized cell number in the gut to the number of cells infected by the same virus within pharyngeal junction anterior to foregut (An.foregut). We found that RET perturbation resulted in 86% reduction of cells in the foregut, as compared with the control (page 14 line 16-17).

2. *Is the inhibition of CXCR4/RET affecting neural crest migration, survival or both?*

We now clarify that inhibition of CXCR4 or RET signaling appears to be selectively affecting direction of neural crest migration. We do not see a diminution of neural crest cells so think that survival is not affected, particularly since the dominant negative constructs are only synthesized at a high level after 1 day, such that the cells are well on their way to the branchial arches prior to onset of the inhibitory effect. Based on the quantification, similar number of dominant negative virus infected cells were present in the cranial nerve and pharyngeal junction when compared with control virus. However, there is clearly a major effect on the direction that the cells pursue once in the branchial arches (page 17 line 5-10).

3. *Is the proportion of cardiac neural crest that contribute to the gut affected when CXCR4 is inhibited?*

At E7, the ratio between cell number in the gut and cranial ganglia is similar between CXCR4-perturbed cells (13.3%) and control cells (13.9%), suggesting that inhibition of CXCR4 signaling does not change the numbers of neural crest cells migrating to the gut but rather diverts cells that would normally migrate to the heart to this location (page 14 line 2-7).

Reviewer 5

1. *This new study drills down on these earlier observations by using novel lineage tracing and imaging tools that enable identification of contributions at the single cell level. Nevertheless, this is a minor advance in what is known about contributions to cardiac and enteric neural crest fates.*

We respectfully disagree with the reviewer regarding the level of advance and significance of our findings. All previous attempts to look at multipotency of the cardiac and vagal neural crest were done *in vitro* and therefore prone to tissue culture artifacts. While much more difficult to do *in vivo*, it is critical to revisit these questions in the normal context of the embryo and at the level of single cells and their progeny.

2. *The authors provided an adequate response to the second criticism by the addition to the manuscript of neural crest cell fate markers specific to sympathetic neurons and Schwann cells, and muscle actin as an indicator of mesenchymal cell fate. Further, the authors addressed molecular guidance cues specific to localization of cardiac and enteric neural crest. Their analysis suggests that individual neural crest cells coming from the branchial arch region may be equally capable of contributing to either heart or gut. This is a reasonable interpretation and strengthens the study.*

We thank the reviewer for this input.